# Conformational equilibrium shift underlies altered K+ channel gating as revealed by NMR

Yuta Iwahashi[1], Yuki Toyama[1], Shunsuke Imai [1], Hiroaki Itoh [1], Masanori Osawa [1,3], Masayuki Inoue [1] & Ichio Shimada [1,2 ✉]

The potassium ion (K+) channel plays a fundamental role in controlling K+ permeation across the cell membrane and regulating cellular excitabilities. Mutations in the transmembrane pore reportedly affect the gating transitions of K+ channels, and are associated with the onset of neural disorders. However, due to the lack of structural and dynamic insights into the functions of K+ channels, the structural mechanism by which these mutations cause K+ channel dysfunctions remains elusive. Here, we used nuclear magnetic resonance spectroscopy to investigate the structural mechanism underlying the decreased K+-permeation caused by disease-related mutations, using the prokaryotic K+ channel KcsA. We demonstrated that the conformational equilibrium in the transmembrane region is shifted toward the non-conductive state with the closed intracellular K+-gate in the disease-related mutant. We also demonstrated that this equilibrium shift is attributable to the additional steric contacts in the open-conductive structure, which are evoked by the increased side-chain bulkiness of the residues lining the transmembrane helix. Our results suggest that the alteration in the conformational equilibrium of the intracellular K+-gate is one of the fundamental mechanisms underlying the dysfunctions of K+ channels caused by disease-related mutations.

[1] Graduate School of Pharmaceutical Sciences, The University of Tokyo, Hongo, Bunkyo-ku, Tokyo 113-0033, Japan. [2] RIKEN Center for Biosystems Dynamics Research, Kanagawa 230-0045, Japan. [3] Present address: Keio University Faculty of Pharmacy, Shibakoen, Minato-ku, Tokyo 105-8512, Japan. ✉email: ichio.shimada@riken.jp

The potassium ion ($K^+$) channel is a membrane protein that selectively and rapidly permeates $K^+$ across the cell membrane and plays critical roles in regulating cellular excitability[1]. $K^+$ channel gating is strictly regulated in response to changes in the membrane potential, the binding of ligands and regulatory proteins, and other environmental factors (temperature, osmotic stress, lipid compositions, and so on). The alterations in the gating behavior of $K^+$ channels are closely related to the onset of different types of diseases, as evidenced by the fact that various missense mutations of $K^+$ channels have been identified in patients with diseases such as episodic ataxia[2,3], long-QT syndrome[4], and epilepsy[5]. Therefore, it is important to understand the molecular mechanism by which these disease mutations cause the $K^+$ channel dysfunctions at the atomic level, as it would facilitate the design of $K^+$ channel modulators that could have therapeutic utility in disease treatment[6,7].

Structural studies of $K^+$ channels have revealed that the architecture of the pore region forming the $K^+$-pathway is highly conserved[8,9]. The pore region is a tetramer with fourfold symmetry, and the $K^+$-pathway is formed at the center of the tetramer (Supplementary Fig. 1a). There are two gates on the $K^+$-pathway, the extracellular selectivity filter (SF) gate that forms $K^+$-coordination sites, and the intracellular helix bundle crossing (HBC) gate. The $K^+$-permeation is regulated by the allosteric coupling between these two gates[10–13]. Mutational studies using the Shaker voltage-gated $K^+$ (Kv) channel have revealed a specific site in the HBC gate, Val478, hereafter referred to as a "hot spot", where replacements with amino acids with increased side-chain volumes substantially decrease the $K^+$-permeability[14,15]. Interestingly, the same position of the human Kv1.1 channel, Val408, is the representative position of the disease mutation associated with episodic ataxia[2,16]. The V408A mutation of Kv1.1 decreases the open time duration at the single-channel level[17], and heterozygous mice with the V408A mutation in one Kv1.1 allele have provided an efficient animal model of episodic ataxia type-1[18]. Although bulky or compact amino acids in this hot spot position decrease the $K^+$-permeation in different manners between the Kv families, understanding how these hot spot mutations affect the structure of the $K^+$ channel gates would provide useful insights into the mechanisms of $K^+$ channel dysfunctions caused by disease mutations. The underlying mechanism of the altered $K^+$-permeation in the mutant has been explored by a set of electrophysiological studies. In the case of the V478W mutant of the Shaker Kv channel, the opening of the HBC gate is hampered, as evidenced by the loss of the effect of an intracellular blocker[15]. Lu and co-workers solved the crystal structure of a Kv1.2-Kv2.1 chimera with a Val to Trp mutation at the corresponding position (V406W)[19]. Although the mutant structure exhibited reduced $K^+$-density at the SF gate, providing structural insight into the macroscopic inactivation process, the overall structure of the pore region, including the HBC gate, showed few structural differences between the wild-type and mutant proteins, with a root mean square deviation of 0.3 Å. Therefore, the means by which this hot spot mutation affects the transmembrane pore structure still remain elusive. In order to address this situation, the dynamic aspects of $K^+$ channels must be analyzed, because various spectroscopic studies have demonstrated that the $K^+$ channel proteins are highly plastic and their dynamic behavior is essential for the $K^+$ channel activity[20–24]. These inherently dynamic properties of $K^+$ channels can be overlooked by crystal structural studies because the crystal structure usually represents the stably-formed, ground-state conformation.

Here, we investigated the structural mechanism underlying the altered gating behavior of $K^+$ channels induced by the hot spot mutation, by solution nuclear magnetic resonance (NMR) spectroscopy, which can characterize the conformational dynamics of

$K^+$ channels in a physiologically relevant solution environment. Using a prototypical $K^+$ channel from *Streptomyces lividans*, KcsA, we demonstrated that the Ala to Val mutation at the hot spot residue greatly stabilizes the structure with the closed HBC gate, by affecting the conformational exchange process accompanying its structural rearrangements. Furthermore, we identified a set of mutants in which the conformational exchange process is altered to different extents. The NMR characterizations of the exchange process, along with the single-channel analyses of these mutants, allowed us to clarify the structural mechanism of the multi-timescale gating behavior of KcsA. Our results show that the change in the conformational dynamics of the HBC gate underlies the altered gating behavior observed in the disease-related mutants of $K^+$ channels.

## Results

**A111V mutant of KcsA adopts a closed HBC gate structure**. In this study, we used the pH-dependent $K^+$ channel from *Streptomyces lividans*, KcsA, to investigate the gating mechanisms of the Kv channels[25]. KcsA shows similar electrophysiological properties to those observed in human Kv channels, and thus serves as a prototypical counterpart of eukaryotic Kv channels[26,27]. Moreover, KcsA lacks a voltage-sensor domain and is composed solely of the transmembrane pore region, highlighting its suitability for investigating the effects of the hot spot mutation on the structure and dynamics of the pore region. KcsA is a homo-tetrameric channel, and each subunit is composed of three transmembrane helices, referred to as the outer helix, pore helix, and inner helix, forming the transmembrane $K^+$-pore, followed by the C-terminal intracellular region that stabilizes the tetrameric structure and plays a modulatory role in the pH-dependent gating[8,28–30]. Our previous NMR studies revealed that the structural changes of the HBC gate and the SF gate are tightly coupled, and KcsA adopts three distinct conformational states in response to changes in pH, $K^+$-concentration, and temperature (Supplementary Fig. 1a)[10,30,31]. At pH 6.5, the pH-sensor residue His25 is deprotonated and KcsA adopts the closed (C) state with the closed HBC gate (Supplementary Fig. 1b). When the pH drops to 3.0 and His25 is protonated, the HBC gate adopts the open structure and KcsA exists in an equilibrium between permeable (P) and impermeable (I) states, which differ in the structure of the SF gate (Supplementary Fig. 1c). The relative populations are sensitive to changes in $K^+$-concentration and temperature, and the population of the I state increases as the temperature or $K^+$-concentration decreases. This three-state model is highly consistent with the gating cycle modeled from the X-ray crystallographic[11,32] and solid-state NMR studies[13,21]: the C, P, and I states correspond to the Closed (C)/Open (O), O/O, and O/Inactivated (I) states of the HBC and SF gates, respectively. The NMR signals from the methyl groups of Leu59δ1 and Val76γ1 showed different chemical shifts between the three states, and therefore these signals serve as a fingerprint of the conformational states with the different structures of the SF and HBC gates.

The sequence alignment of the KcsA and Kv channels revealed that the hot spot residue in KcsA is Ala111, which corresponds to Val408 in Kv1.1 and Val478 in Shaker Kv (Fig. 1a). Kitaguchi et al. reported that the gating behavior in Shaker Kv channels was highly variable, depending on the volume of the side-chain at this position[15], and thus we replaced Ala at position 111 with Val and characterized the structure of the mutant KcsA. We compared the $^1H$-$^{13}C$ heteronuclear multiple quantum coherence (HMQC) spectra of the wild-type and the A111V mutant solubilized in *n*-dodecyl-β-D-maltopyranoside (DDM) at pH 3.0 and 45 °C in the presence of 100 mM KCl, using the uniformly deuterated and

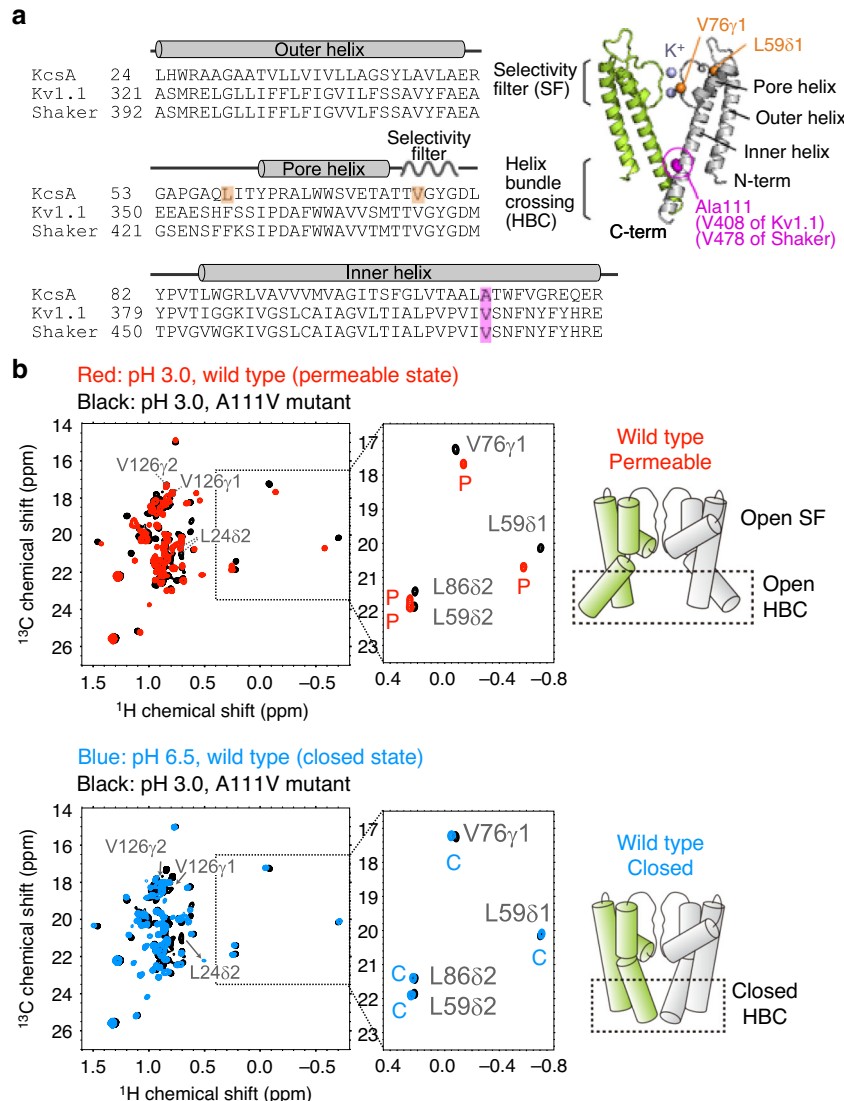

**Fig. 1 The hot spot mutation in the transmembrane region and the NMR analyses of the A111V mutant of KcsA. a** Sequence alignment of *Streptomyces lividans* KcsA (Uniprot ID: P0A334), *Homo sapiens* Kv1.1 (Uniprot ID: Q09470), and *Drosophila melanogaster* Shaker Kv channel (Uniprot ID: P08510). The hot spot positions in the inner transmembrane helix are highlighted in magenta. The hot spot position of KcsA, Ala111, is indicated in the crystal structure of KcsA (PDB ID: 1K4C). Only two-facing subunits are shown for clarity. The positions of the fingerprint methyl groups, Val76γ1 and Leu59δ1, are highlighted in orange. **b** Overlay of the $^1$H-$^{13}$C HMQC spectra of the wild type at pH 3.0 (red) and the A111V mutant at pH 3.0 (black) (top), and overlay of the spectra of the wild type at pH 6.5 (blue) and the A111V mutant at pH 3.0 (black) (bottom). The spectra were measured at 45 °C and 18.8 Tesla (800 MHz $^1$H frequency), in the presence of 100 mM KCl. The chemical shifts of Leu24 and Val126 in the A111V mutant at pH 3.0 were different from those of the wild type at pH 6.5, because these residues are located proximate to the protonatable residues, His25, Glu118, Glu120, Arg127, and His128, and thereby reflect the local structural differences accompanied by the protonation of KcsA at pH 3.0. Schematic models of the permeable (P) and closed (C) states are shown on the right.

Ileδ1, Leuδ1/δ2, and Valγ1/γ2 selectively methyl-$^1$H$_3$-$^{13}$C labeled samples (Fig. 1b and Supplementary Fig. 2). Under these conditions, the wild-type KcsA adopts the P state, in which both the SF and HBC gates form the open structures[10]. In the A111V mutant, the chemical shifts of the fingerprint signals (Leu59δ1 and Val76γ1 methyl) were different from those of the wild type, indicating that the A111V mutant adopts a different conformational state from that of the wild type at pH 3.0. Notably, the chemical shifts of the Leu59δ1 and Val76γ1 methyl groups matched those of the wild type at pH 6.5, in which the wild-type KcsA adopts the C state. This result indicates that the A111V mutant forms the structure that closely resembles the C state in the wild type, even under pH 3.0 condition. The chemical shifts of the other methyl groups, such as Leu59δ2 and Leu86δ2, also

matched those of the C state of the wild type, indicating that the overall transmembrane pore structure of the A111V mutant adopts the structure of the C state observed in the wild type at pH 6.5.

Interestingly, some methyl signals did not follow this trend. These exceptional methyl groups are located at the cytoplasmic end of the outer transmembrane helix (Leu24) and in the C-terminal helix bundle region (Val126, Leu151, and Leu155), and exhibited the same or similar chemical shifts as those of the wild type at pH 3.0 (red stars, Supplementary Fig. 3). In our previous pH titration analyses of KcsA in DDM micelles, we demonstrated that the C-terminal region exhibited a pH-dependent structural transition (pH$_{1/2}$ = 6.4 ± 0.4) that was not coupled to the pH-dependent gating transition at the transmembrane pore (pH$_{1/2}$ =

$5.0 \pm 0.3$)[10]. In addition, Leu24 is located next to the pH-sensor residue, His25, and is expected to be sensitive to the perturbation of the salt-bridge network formed by His25 under acidic conditions[31,33]. These results indicate that the A111V mutant is likely to be protonated at pH 3.0, which caused the pH-dependent structural transitions at the C-terminal region and the perturbation of the salt-bridge network centered on His25, but could not induce the structural changes of the transmembrane pore, and thus the pore region is forced to adopt the C state conformation, regardless of the pH condition. As the C state represents the nonconductive structure, in which the $K^+$-permeation is blocked by the constricted HBC gate, our results indicate that the A111V mutant is a nonconductive mutant whose HBC gate adopts the closed structure even under acidic conditions. It was difficult to directly observe the residues forming the constriction point of the HBC $K^+$-gate and obtain the distance information of the gate structure, due to the signal overlapping and broadening observed in Leu105, Leu110, and Val115, which presumably reflect the conformational heterogeneity of the salt-bridge network formed between His25, Glu118, Glu120, Arg121, and Arg122, as proposed by Thompson et al.[33], and/or additional local conformational exchange processes associated with it. However, the structural changes in the transmembrane pore region are quite highly cooperative via an extensive allosteric network formed by Thr74, Ile100, and Phe103[11-13], and the mutation site, Ala111, is distant from these residues (>13 Å in the Cβ-Cβ distance) and not directly involved in this network. Therefore, the chemical shift pattern of the fingerprint methyl groups in the A111V mutant most likely reflects the formation of a fully or at least partially closed HBC structure.

Although this trend is in contrast to the case observed with the V408A mutant of the Kv1.1 channel, where the introduction of a more compact amino acid decreases the $K^+$-permeation, the results obtained with KcsA are consistent with the case of the Shaker Kv channel, in that the bulky side-chain at the corresponding hot spot position results in a nonconductive phenotype[15]. Thus, the A111V mutant of KcsA can serve as a useful model for examining the effects of hot spot mutations in Shaker-like Kv channels.

**Differences in steric contacts affect the conformational equilibrium of KcsA.** To further investigate the mechanism by which the bulkiness of the side-chain in the transmembrane region affects the gating of KcsA, we systematically mutated the methyl-containing residues in the transmembrane region to those with different side-chain volumes, as mutations of the methyl-containing residues were proposed to be useful for investigating an allosteric communication at various locations throughout the protein[34,35]. We observed HMQC spectra of 25 mutants (V34I, L35I, L36I, V37I, V39I, L40I, L41I, L46I, V48I, L49I, L66I, V70I, L81I, V84I, L86I, L90I, V91I, V93I, V94I, V95I, V97I, L105I, V106I, L110I, and V115I) at pH 3.0 and 45 °C in the presence of 100 mM KCl, and compared the conformational states of these mutants with that of the wild type. Among these mutants, we observed a marked difference in the conformational states in the L36I, L46I, V91I, and V106I mutants (Fig. 2a and Supplementary Fig. 2). In the L36I and V106I mutants, the chemical shifts of Leu59δ1 and Val76γ1 matched those of the C state, indicating that these mutants mainly adopt the C state under the pH 3.0 and 45 °C conditions, as similarly observed in the A111V mutant. In the L46I and V91I mutants, the signals of Leu59δ1 and Val76γ1 split into two signals, and the chemical shifts of these split signals matched those of the C and P states, indicating that these mutants exist in a conformational equilibrium between the

C and P states under these conditions (Fig. 2b and Supplementary Fig. 2).

We then compared the temperature-dependent shift in the equilibrium of these mutants with that of the wild type (Fig. 2c). In the wild type, the population of the P state decreases and the equilibrium is shifted toward the I state as the temperature decreases, as previously reported[10]. In the V91I and V106I mutants, we observed a set of split signals with chemical shifts that matched those of the three states (the P, C, and I states) in the wild type. Interestingly, the signals from all three of the states in these mutants were observed below 40 °C, and their relative populations changed in a temperature-dependent manner. These results suggest that these KcsA mutants exist in a similar conformational equilibrium between the three states to that observed in the wild type, and the C state is more stabilized with the V91I and V106I mutations under the acidic conditions, as compared to the wild type. The observed temperature-dependent spectral changes also indicated that the exchange kinetics between these states was different between the mutants. In the V91I mutant, the signals from all three states were separately observed, while in the V106I mutant, the signals from the P and C states were merged to give a single broad peak at temperatures above 35 °C, strongly suggesting that the exchange rate between the P and C states is faster in the V106I mutant than in the V91I mutant. Notably, in the A111V mutant, the P and I states were not observed between 25 and 45 °C, demonstrating that the introduction of the bulky amino acid at the hot spot position greatly alters the conformational equilibrium and forces KcsA to adopt exclusively the C state with the closed HBC gate.

The methyl-containing residues with side-chain bulkiness that affected the three-state conformational equilibrium identified in this study were clustered in three regions: the extracellular-side (Leu46 and Val91), mid-transmembrane (Leu36 and Val106), and intracellular-side (Ala111) regions (Fig. 2a). We then compared the side-chain interactions of these residues between the HBC-closed and open crystal structures. We found that the mid-transmembrane and intracellular-side residues form extensive inter-transmembrane helix van-der-Waals contacts in the HBC-open structure (6.1 Å between Leu36Cβ and Val106Cβ; 5.4 Å between Leu105Cβ and Ala111Cβ in the open structure, as compared to 10.5 Å between Leu36Cβ and Val106Cβ; 9.0 Å between Leu105Cβ and Ala111Cβ in the closed structure) (Fig. 3a, b)[32,36]. Together with the fact that the C state is relatively stabilized by introducing mutations with increased side-chain volumes (L36I, L46I, V91I, V106I, and A111V), we considered that the disfavored side-chain van-der-Waals contacts in the open HBC structure to be the underlying mechanism of the increased C state population in these mutants. To further support this concept, we designed two additional mutants, A32V and T33I, in which Ala32 and Thr33 were mutated to a bulkier valine or isoleucine residue. The side-chains of Ala32 and Thr33 are closer to the contacting residues in the HBC-open structure than in the HBC-closed structure. Therefore, the introduction of bulky amino acids at these positions is expected to destabilize the HBC-open structure and hence should also shift the equilibrium to the C state (3.8 Å between Ala32Cβ and Leu105Cβ; 5.5 Å between Thr33Cβ and V106Cβ in the open structure, as compared to 4.7 Å between Ala32Cβ and Leu105Cβ; 8.1 Å between Thr33Cβ and Val106Cβ in the closed structure). We also mutated Ala28 to valine, as its side-chain is farther from the mid-transmembrane cluster in the HBC-open structure, to serve as a negative control (7.8 Å between Ala28Cβ and Leu105Cβ in the open structure, as compared to 6.1 Å between Ala28Cβ and Leu105Cβ in the closed structure) (Fig. 3b). We compared the $^1H$-$^{13}C$ HMQC spectra of these mutations with that of the wild

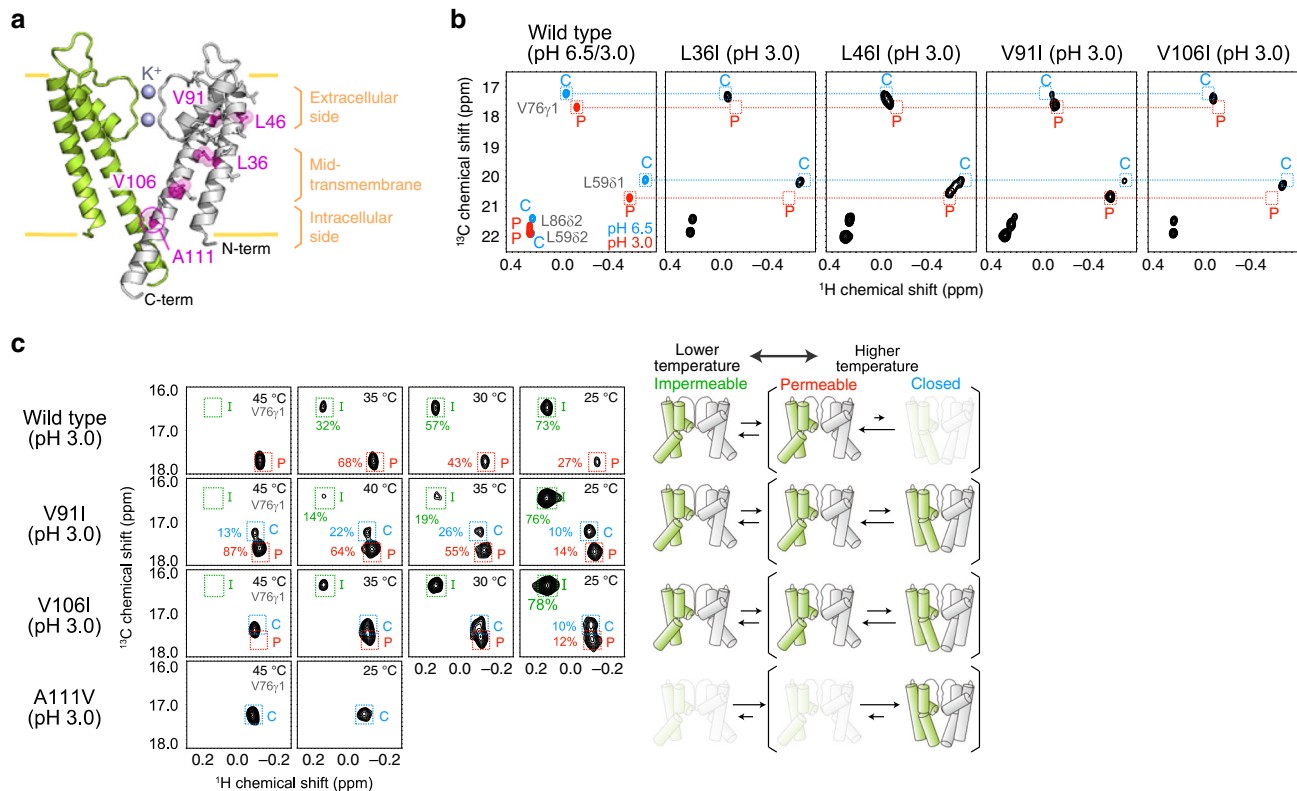

**Fig. 2 NMR analyses of the KcsA mutants. a** Mapping of the mutated Ile, Leu, and Val residues on the crystal structure of KcsA (PDB ID: 1K4C). Ile, Leu, and Val residues are shown as stick models in one of the subunits. The residues with mutations that affect the conformational equilibrium between the permeable (P) and closed (C) states are highlighted in magenta. Only two-facing subunits are shown for clarity. The transmembrane region is schematically represented with yellow lines and three clusters of the mutated residues are indicated on the right. **b** $^1$H-$^{13}$C HMQC spectra of the wild type and the mutants. The regions for the fingerprint residues, Val76 and Leu59, are shown. The chemical shifts of the P state (pH 3.0, 45 °C) and the C state (pH 6.5, 45 °C) in the wild type are indicated as red and blue boxes, respectively. The spectra were measured at 45 °C in the presence of 100 mM KCl. **c** Val76γ1 methyl signals of the wild type and the V91I, V106I, and A111V mutants, measured at different temperatures. The chemical shifts of the I state (pH 3.0, 25 °C), the P state (pH 3.0, 45 °C), and the C state (pH 6.5, 45 °C) in the wild type are indicated in green, red, and blue boxes, respectively. Schematic models of the equilibrium shifts in the mutants are shown on the right. The spectra were recorded at 11.7 Tesla (500 MHz $^1$H frequency, for L36I), 14.1 Tesla (600 MHz $^1$H frequency, for L46I), or 18.8 Tesla (800 MHz $^1$H frequency, for wild type, V91I, V106I, and A111V).

type (Fig. 3c and Supplementary Fig. 2). As expected, the relative population of the C state was markedly increased in the A32V and T33I mutants, while the A28V mutation did not affect the conformational equilibrium of KcsA. These results further support our proposal that the alteration in the van-der-Waals contact is the underlying mechanism for the shift in the conformational equilibrium of KcsA.

**Conformational equilibrium between the P and C states is responsible for intra-burst gating.** We next compared the K$^+$ channel activities of the wild type and the aforementioned mutants, to investigate how the altered conformational equilibrium between the P and C states affects the single-channel behavior. First, we recorded the single-channel activity of the wild type in a planar lipid bilayer at room temperature. In accordance with the previous report[37–39], the wild-type KcsA exchanged between the short duration of burst periods (Fig. 4a, gray half parenthesis) and the long duration of an inter-burst nonconductive state (Fig. 4a, green half parenthesis) on a 1–10 s timescale, and during the burst period, the wild-type KcsA mainly remained in the conductive state. In the wild-type KcsA, the populations of the conductive and the nonconductive states during the burst period were calculated to be 84% and 16%, respectively (Fig. 4a top, red and blue dotted lines). In the previous reports by Chakrapani et al., three different modal

conducting behaviors were observed during the burst period in the single-channel recordings of the wild-type KcsA: the high-P$_o$ mode with the open probability of 82%, the flicker mode with the open probability of 40%, and the low-P$_o$ mode with the open probability of 16%[38,39]. In our single-channel analyses, only the high-P$_o$ mode was observed in the wild-type recordings, which is consistent with the previous observation that the high-P$_o$ mode is the most prevalent gating mode of the wild-type KcsA. We then compared the single-channel activity of the wild type with that of the V91I mutant, in which the P and C states had similar populations at 25 °C in the NMR experiments (Fig. 2c). Notably, as compared to the wild type, the V91I mutant exhibited significantly lower basal activity, which could be characterized by the frequent transitions between the conductive and nonconductive states during the burst period and the increased population of the nonconductive state during the burst period (Fig. 2a, blue dotted line). The K$^+$-current exhibited a broader distribution than in the wild type as shown in the all-points histogram, suggesting there are multiple sub-conductance levels in the V91I mutant. From the fitting of the distribution of the V91I mutant assuming two states, conducive and K$^+$-nonconductive states, the populations of these states were calculated to be 30% and 70%, respectively (Fig. 4a bottom, red and blue dotted lines), indicating that the relative populations of these two states during the burst period are inverted in the V91I mutant as compared to the wild type. Considering the NMR results that, in the V91I mutant, the

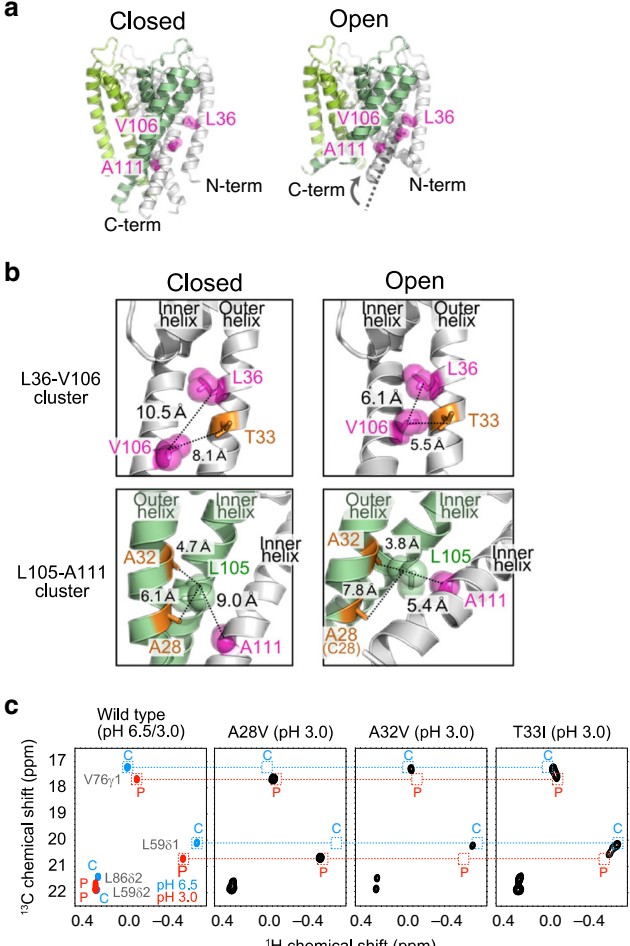

**Fig. 3 Side-chain interactions in the open and closed structures of the HBC gate. a** Crystal structures of KcsA with the closed (PDB ID: 1K4C) and open (PDB ID: 5VK6) HBC gate structures. Each subunit in the tetramer is colored differently (white for the right and back subunits, light green for the left subunit, and green for the front subunit). Leu36, Val106, and Ala111 are highlighted in magenta in one of the subunits. **b** Close-up views of Thr33, Leu36, and Val106 (upper panel) and Ala28, Ala32, Leu105, and Ala111 (lower panel) in the closed (left) and open (right) HBC gate structures. The Leu36-Val106 cluster includes intra-subunit interactions, and the Leu105-Ala111 cluster includes both intra-subunit (within green subunit) and inter-subunit (between green and white subunits) interactions. The residues characterized in Fig. 2 are colored magenta, and the residues replaced with the designed mutations are colored orange. Ala28 is replaced with Cys28 in the crystal structure with the open HBC gate structure. **c** $^1$H-$^{13}$C HMQC spectra of the wild type and the A28V, A32V, and T33I mutants. The regions for the fingerprint residues, Val76 and Leu59, are shown. The chemical shifts of the P state (pH 3.0, 45 °C) and the C state (pH 6.5, 45 °C) in the wild type are indicated in red and blue boxes, respectively. The spectra were measured at 45 °C and 14.1 Tesla (600 MHz $^1$H frequency, for A28V and A32V), or 18.8 Tesla (800 MHz $^1$H frequency, for T33I), in the presence of 100 mM KCl.

population of the C state markedly increased whereas the population of the I state was not largely affected, these results suggest that the intra-burst gating transitions are attributable to the exchange between the P and C states, rather than that between the P and I states (Figs. 2c and 4a).

We also measured the single-channel activities of the V106I and A111V mutants, which showed the increased population of the C state in the NMR analyses. The populations of the

nonconductive state during the burst period were calculated to be 45% in the V106I mutant and more than 90% in the A111V mutant, respectively, indicating the nonconductive populations were similarly increased in these mutants as compared to that in the wild type (16%) (Fig. 4b, blue dotted lines). The broad distribution of the K$^+$-conductive state was similarly observed in the V106I mutant, showing the existence of the multiple sub-conductance levels as observed in the V91I mutant. The relative populations of the C state, normalized by the sum of the populations of the P and C states, correlated well with the nonconductive state populations during the burst period in the single-channel recordings ($R^2 = 0.89$), supporting our notion that the P and C states defined by the NMR analyses correspond to the conductive and nonconductive states during the burst period, respectively, observed in single-channel recordings (Fig. 4c).

Although we could observe a positive correlation between the P and C state populations defined by the NMR analyses and the intra-burst conductive/nonconductive populations, it was difficult to quantitatively determine the population of the long, inter-burst nonconductive state due to the insufficient numbers of sampling points of the slow gating transitions, which hampered the analysis of the correlation including this state. Therefore, we compared the exchange kinetics between the P and C states with the intra-burst gating kinetics, to obtain further supportive data. We analyzed the intra-burst gating kinetics of the V91I mutant and calculated the mean dwell times of the conductive and nonconductive states within the burst period (Supplementary Fig. 4). The mean dwell times of the conductive and nonconductive states were 7.9 and 27.0 ms, respectively, and correspond to an exchange rate of 160 s$^{-1}$ assuming two-state exchange. These kinetic parameters are consistent with the NMR observation that the exchange between the P and C states was in a slow-exchange regime with significant exchange-induced line broadening effects, and cannot be explained by the exchange between the P and I states, as we previously demonstrated that this exchange process is much slower, with a $k_{ex}$ of 1.4 s$^{-1}$ at 40 °C. In the NMR spectrum of the V91I mutant at 25 °C, extra line broadening effects on the order of several tens of seconds were observed for the V76γ1 methyl signals in the $^{13}$C dimension, as compared to the wild type (Supplementary Fig. 5), consistent with the expected exchange contribution (=46.9 s$^{-1}$) assuming the exchange rate of 160 s$^{-1}$, the $^{13}$C chemical shift difference of 0.45 ppm, and the major and minor state populations of 70 and 30% at 18.8 Tesla (800 MHz $^1$H frequency)[40]. These results further support our proposal that the exchange between the P and C states explains the intra-burst conductive and nonconductive transitions.

**Description of the KcsA single-channel gating behavior by the three distinct conformational states defined by NMR.** On the basis of these observations, we modeled the single-channel gating behavior of KcsA under acidic conditions, according to the three distinct conformational states defined by the NMR analyses (Fig. 4d). To date, numerous electrophysiological studies have established that KcsA gating can be modeled by multiple transitions with different time scales, as stated above[37–39,41]: a slow gating between the burst and the inter-burst nonconductive states on a timescale on the order of seconds, and intra-burst gating on a timescale on the order of milliseconds. However, the detailed structural mechanism of this multi-timescale behavior has remained poorly understood. Regarding the slow gating process, we previously proposed that this gating process could be explained by the slow conformational equilibrium between the P and I states, which differ in the structure of the SF gate, and

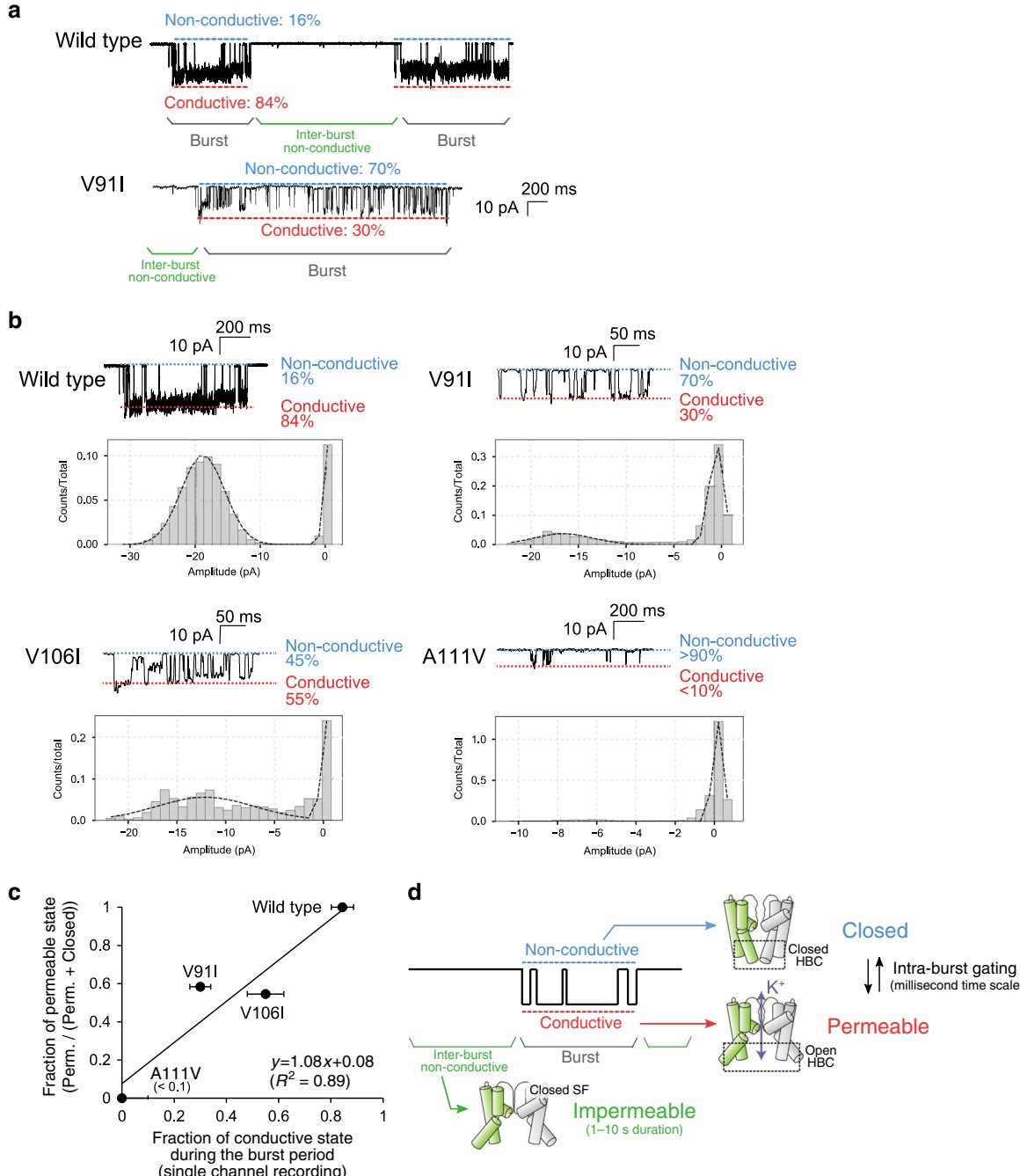

**Fig. 4 Single-channel analyses of KcsA. a** Representative single-channel current traces of the wild type and the V91I mutant of KcsA, obtained with the planar lipid bilayer system. Currents were recorded at −200 mV with a symmetric K$^+$ concentration of 100 mM. The asymmetric pH conditions (pH 6.5/3.0) were used to observe the K$^+$-currents of KcsA, with the intracellular side directed toward the acidic side. The burst period and the inter-burst nonconductive state are donated as gray and green half-parentheses, and the conductive and nonconductive states during the burst period are marked with the red and blue dotted lines, respectively. **b** Open probabilities during the burst period in the wild type and the V91I, V106I, and A111V mutants. All-points histograms of the channel openings from each recording are shown below. The open probabilities were calculated from the deconvolution assuming two Gaussian distributions corresponding to the conductive and nonconductive states (dotted lines). **c** Correlation plot between the fraction of the P state in the NMR analyses and the fraction of the conductive state during the burst period in the single-channel analyses ($R^2 = 0.89$). The centre of the plot represents the fraction of the density of the conductive distribution, and the error bars represent one standard deviation, as estimated from a covariance matrix used to fit the histogram. The number of data points used to fit the Gaussian distributions (the number of bins = 25) was $n = 29,123$ for wild type, $n = 10,474$ for the V91I mutant, and $n = 2,353$ for the V91I mutant. The fractions of the P state are calculated from the signal intensities of the P state divided by the sum of the intensities of the P and C states, using the NMR spectra recorded at 25 °C and pH 3.0, in the presence of 100 mM K$^+$. **d** Schematic model of the single-channel gating behavior described by the three different conformational states of KcsA. Source data are provided as a Source data file.

demonstrated that these two states are exchanging on a timescale on the order of 1 s[10,30]. This notion is further supported by the results obtained with the E71A and Y82A mutants, in which the P and I states are dominantly populated in our NMR analyses, respectively[10,30]. Given the electrophysiological results that the E71A mutant does not exhibit the long closure and the Y82A mutant exhibits the prolonged closure interspersed with the shorter burst period[32,37,39], we considered that the nonconductive state between the burst periods (Fig. 4d, green), of which duration time is on the second timescale, corresponds to the I state defined by the NMR analyses.

Regarding the fast, intra-burst gating of KcsA, the structural mechanism is still under debate, and there are several proposed mechanisms in which the gating is described by the conformational transitions of the SF gate or the HBC gate[20,37,39]. Here, during the course of the characterizations of the disease-related A111V mutant, we identified a set of novel mutants in which the intra-burst gating is altered, and showed that the nonconductive state population within the burst period correlates with the C state population. Furthermore, the kinetics of the exchange between the P state and the C states, estimated from the NMR line-shapes, was also consistent with the intra-burst gating kinetics in the single-channel recordings (Supplementary Fig. 5). These results indicate that the millisecond-order gating transition between the conductive (Fig. 4d, red) and nonconductive states (Fig. 4d, blue) during the burst period corresponds to the exchange between the P and C states defined by the NMR analyses. This model describes the single-channel behavior of KcsA with two different nonconductive conformational states, the I state with the closed SF gate and the C state with the closed HBC gate, and the highly variable single-channel kinetic behavior can be explained by the different lifetimes of these two conformational states.

## Discussion

In this study, we investigated the structural mechanism underlying the impaired K$^+$ channel activity induced by the hot spot mutation. We demonstrated that altered steric contacts in the transmembrane region dramatically affect the conformational equilibrium in the transmembrane HBC gate. We also revealed that KcsA has a similar property to that of the Shaker K$^+$ channel and that the introduction of the bulky amino acid in the hot spot region shifts the equilibrium to stabilize the HBC-closed structure, thus leading to the low K$^+$-permeation in the mutant. This structural model is based on our NMR analyses of the KcsA mutants that were conducted in micellar environments, even though the KcsA channel activity is sensitive to the surrounding membrane environments[42–45]. However, we previously demonstrated that the three distinct conformational substates of KcsA (P, C, and I states) were well preserved in the native membrane environments, based on NMR analyses of KcsA reconstituted into high-density lipoprotein[30]. Therefore, it would be reasonable to compare the energetic contribution of each mutation to the conformational landscape of KcsA and expect that the mutations can alter the conformational equilibrium of KcsA in native membrane environments as observed in the DDM micelles.

It should be noted that the stabilization of the HBC-closed structure by bulky or compact side-chains is different between the Kv families, and we presume that these differences are attributable to the variations of the transmembrane residues forming intrahelical contacts and/or the possible small differences in the orientation of the transmembrane helices. In the case of the disease-related V408A mutant of Kv1.1, the side-chain of Val408 probably forms the unfavored steric contacts in the conformational state with the closed HBC gate, and thus the introduction

of the V408A mutation would shift the equilibrium to stabilize the structure of the closed HBC gate and hence decrease K$^+$-permeation. The alterations in the steric contacts in the transmembrane region can also be related to the diseases associated with different Kv families, as well as the Shaker-like Kv1 family focused in this study. Missense mutations in the transmembrane helix of Kv7.1 (KCNQ1) are found in patients with long-QT syndrome, and some mutations, such as Ala341 (Gly104 in KcsA) to Val and Leu342 (Leu105 in KcsA) to Phe, have increased side-chain volumes in the region around the hot spot residues[46–48]. Thus, the mechanism proposed here could be generally applicable to some of the other diseases associated with K$^+$ channel mutations.

The alterations in the steric contacts in the transmembrane region are also related to the fine-tuning of the K$^+$ channel activity under physiological conditions. First, the RNA editing of Kv1, which transforms the genomic form Ile321 to the edited form Val321, is reportedly responsible for cold environment adaptation by stabilizing the nonconductive state[49]. The edited residue, Ile/Val321 of Kv1, corresponds to Thr33 of KcsA and is located close to the hot spot region, suggesting that the K$^+$ channel activities are fine-tuned by RNA editing, which alters the steric contacts in the hot spot region without dramatically affecting the overall channel architecture. Second, the K$^+$ channel activity is controlled by lipidic molecules[30,42–45]. The transmembrane clusters identified in our study are exposed to the membrane lipids, for example, the extracellular-side cluster formed by Leu46 and Val91 is a lipid-binding cleft identified in the crystal structure of the Kv1.2-Kv2.1 chimera[50], and the residue in the intracellular cluster, Leu105, is also oriented toward the lipid membrane. Thus, we propose that the different steric contacts between the lipid acyl chains and these hydrophobic residues are the underlying mechanism of the modulations of K$^+$ channel gating by lipid molecules.

By combining the NMR analyses of the conformational equilibrium and the single-channel analyses, we modeled the single-channel gating transition by the three distinct conformational states and showed that the conformational equilibrium in the transmembrane HBC gate is responsible for the intra-burst gating behavior. So far, the structural transition of the SF gate has been considered to be mainly responsible for the gating under acidic conditions[21,27,32,51]. Here, we showed that the HBC gate can also function as the main gate under acidic conditions, and the combination of the SF and HBC structural changes aptly explains the complex single-channel gating of KcsA composed of multiple nonconductive states with different dwell times; i.e., the 1–10 second-order long closure interspersed between bursts and the millisecond-order short closure during the burst[37–39]. This model implies that the exchange kinetics between the permeable and impermeable states, which results from the structural rearrangement of the SF gate, is slower than that between the closed and permeable states, which results from the rearrangement of the HBC gate. These exchange kinetics profiles are in good agreement with the results of the electrophysiological measurements of the KcsA macroscopic currents, in which the pH-dependent activation or deactivation process, involving the structural changes at the HBC gate, occurs on a 10–10$^2$ ms timescale, whereas the subsequent inactivation process involving the SF gate closure occurs on a 1–10 s timescale[27,32,41,51]. The apparently slower timescale for the SF gating is attributed to the fact that the closed SF gate is highly stabilized by the extensive hydrogen-bond networks formed by the water molecules bound behind the SF gate, leading to the higher activation free energy barrier for the structural rearrangements of the SF gate. These bound-water molecules were first identified in the high-resolution crystal structure[36], and were also experimentally observed in our

solution NMR and the solid-state NMR studies[10,52]. Roux and co-workers revealed that the SF gate opening is controlled by these water molecules and the free energy barrier was estimated to be on the order of several tens of kcal/mol from the potential of mean force calculations, which reasonably explains the very slow structural transition of the SF gate with the timescale of tens of second[53]. We do not fully exclude the possibility that the SF gate also functions as a gate in the short intra-burst gating in combination with the HBC gate because the mutation of the SF gate residue is also known to affect the alternative mode of the intra-burst gating, called "modal-gating"[38,39]. Recently, Weingarth and co-workers have reported that the dynamics of the SF residues are markedly altered in the Glu71 mutants, which show different modal-gating behaviors[54].

Finally, in the single-channel analyses of the V91I and V106I mutants, we observed multiple sub-conductance levels during the burst gating (Fig. 4b). These multiple conductance levels may be attributable to the conformational heterogeneity of the HBC structure in the mutants, because the overall HBC gate can form multiple asymmetric structures, depending on the number of subunits adopting the open conformation among the 4 subunits in the tetramer, assuming that the 4 subunits in the tetramer are not fully cooperative in the open-closed structural transition. The functional consequences of the asymmetric structures were characterized for the KcsA and KcV channels, by using a hetero-tetrameric protein containing different combinations of wild type and mutant subunits[55,56], and various conductance levels were observed between the hetero-tetrameric proteins with different compositions. Although these analyses focused on the asymmetry of the SF gate, the asymmetry of the HBC gate could also cause multiple sub-conductance levels.

In summary, we demonstrated that a mutation in the trans-membrane hot spot region alters the conformational equilibrium in the HBC gate by perturbing the steric contacts formed between the transmembrane helices, thus leading to the impaired $K^+$-permeation in the mutant. Considering the high conservation of the pore architectures among $K^+$ channels, the proposed mechanism can be generally applied to a diverse set of eukaryotic $K^+$ channels and provides a molecular basis for establishing therapeutic strategies to address neural diseases caused by $K^+$ channel dysfunctions.

## Methods

**Protein expression and purification**. The expression vector contains the gene encoding the wild-type KcsA (residues 1–160) with optimal codon usage for *Escherichia coli* (Supplementary Table 1). An additional sequence, including the decahistidine tag and PreScission™ protease (Cytiva) recognition site, was added immediately before the initial methionine residue. The resultant gene was transferred into the pET24d vector (MilliporeSigma) using the NcoI and SalI restriction sites[10,30,31]. In this study, the C-terminal region of KcsA was not cleaved by chymotrypsin, and thus the full-length KcsA protein was analyzed. All KcsA mutants were constructed with a QuikChange® Site-directed Mutagenesis Kit (Agilent Technologies) using the primers listed in Supplementary Table 1. *E. coli* C41 (DE3) cells (Lucigen), transformed with the plasmid encoding KcsA, were grown at 37 °C in LB media for preparing non-labeled samples, or in deuterated M9 media for preparing isotopically labeled NMR samples. For the selective $^{13}CH_3$-labeling of methyl groups, 50 mg/L of [methyl-$^{13}$C, 3,3-$^2H_2$]-α-ketobutyric acid (for Ileδ1) (Cambridge Isotope Laboratories, Inc.) and 80 mg/L of [3-methyl-$^{13}$C, 3,4,4,4-$^2H_4$]-α-ketoisovaleric acid (for Leuδ1, Leuδ2, Valγ1, and Valγ2) (Cambridge Isotope Laboratories, Inc.) were added 1 h prior to the induction[57]. To obtain the stereospecific assignments for the Leuδ1/δ2 and Valγ1/γ2 signals, stereoselective $^{13}CH_3$-labeling was performed by using [2-$^{13}$C, 4,4,4-$^2H_3$] acetolactate (NMR-Bio) as a precursor[58]. The production of the KcsA protein was induced with 1 mM isopropyl β-D-1-thiogalactopyranoside, at 37 °C for 12 h. The KcsA protein was solubilized in buffer containing 50 mM Tris-HCl pH 8.0, 150 mM NaCl, 150 mM KCl, and 20 mM DDM (Dojindo), and purified to homogeneity by chromatography on HIS-Select Nickel Affinity Gel (Sigma-Aldrich) using buffer containing 50 mM Tris-HCl pH 8.0, 150 mM NaCl, 150 mM KCl, 0.5–5 mM DDM, and 0–300 mM imidazole. The His-tags were with HRV-3C protease (Novagen), and the cleaved His-tags and the protease were removed on HIS-Select Nickel

Affinity Gel in buffer containing 20 mM potassium phosphate pH 6.5, 100 mM KCl, 0.5 mM DDM, and 0–30 mM imidazole. NMR samples were prepared by changing the buffer to 20 mM potassium phosphate, 100 mM KCl, by sequential dilution and concentration with an Amicon® Ultra Centrifuge Filter Unit, NMWL 50 kDa (Merck Millipore). All NMR spectra were recorded in buffer containing 20 mM potassium phosphate (pH 3.0 or 6.5), 100 mM KCl, and ~5 mM DDM in 100% $D_2O$. The pH values in $D_2O$ were calibrated by adding 0.4 pH unit to the reading on the pH meter[59].

**NMR analyses**. All experiments were performed on Bruker Avance 500, 600, or 800 spectrometers equipped with cryogenic probes. All spectra were processed by the Bruker TopSpin 2.1 or 3.1 software, and the data were analyzed and visualized using UCSF Sparky version 3.114 (T. D. Goddard and D. G. Kneller, Sparky 3, University of California, San Francisco, CA). $^1$H-$^{13}$C HMQC spectra were acquired by using the phase-sensitive gradient enhanced-2D HMQC sequence (hmqcetgp in the standard Bruker pulse sequence library)[10,30]. The assignments were established by site-directed mutagenesis using the wild-type KcsA protein[10], and the assignments were transferred to the KcsA mutants. Briefly, all Leu, Val, and Ile residues in KcsA were mutated to Ile (for Leu/Val) or Val (for Ile) one by one, and the $^1$H-$^{13}$C HMQC spectra of each mutant were compared to the wild-type spectra. The stereospecific assignments were obtained by recording the $^1$H-$^{13}$C HMQC spectrum of the uniformly deuterated and selectively Leuδ2 and Valγ2 methyl-$^1H_3$-$^{13}$C labeled sample[58].

**Single-channel analyses**. Giant unilamellar vesicles (GUVs) were prepared by the electroformation methods using a Nanion Vesicle Prep Pro setup (Nanion Technologies). A 5 mg mL$^{-1}$ chloroform solution of diphytanoyl phosphatidylcholine (Avanti Polar Lipids) was placed on the glass surface, and 300 μL of 1 M sorbitol was added to hydrate the lipid film. An alternating electrical voltage of −60 mV was then applied at room temperature for 2 h to form GUVs. The single-channel analyses of KcsA were performed by using a Port-a-Patch system (Nanion Technologies). The planar lipid bilayers were formed by adding GUVs to the chamber under negative pressure, and 1 μL of DDM-solubilized KcsA (50 μM) was directly added to the bilayer for reconstitution. The internal and external solutions contained 20 mM potassium phosphate (pH 6.5/3.0), 100 mM KCl, and the constant transmembrane voltage was set to −200 mV. All traces were recorded with a 40 kHz sampling rate at room temperature (around 18 °C). Multiple gating events were analyzed for the wild-type and the mutant KcsA proteins ($n = 2$–$3$), except for the A111V mutant in which the opening event was rarely observed. Data were analyzed with QuB (https://qub.mandelics.com/)[60].

**Reporting summary**. Further information on research design is available in the Nature Research Reporting Summary linked to this article.

## Data availability

Sequence information on *Streptomyces lividans* KcsA, *Homo sapiens* Kv1.1, and *Drosophila melanogaster* Shaker Kv channels are available in the UniProt Knowledgebase under accession codes P0A334, Q09470, and P08510. The PDB accession codes 1K4C, 3EFF[61], and 5VK6 were used in this study. All other data are available from the corresponding author upon reasonable request. Source data are provided with this paper.

## Code availability

Codes used to fit the single-channel all-points histograms and NMR line-shapes are available from the corresponding author upon request.

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

## Acknowledgements

This work was supported in part by the Japan Agency for Medical Research and Development (AMED) under Grant Number JP18ae0101046 (to I.S.) and the Ministry of Education, Culture, Sports, Science and Technology (MEXT)/Japan Society for the Promotion of Science KAKENHI, Grant Number JP17H06097 (to I.S.).

## Author contributions

Y.I., Y.T., and S.I. purified the proteins and performed the NMR experiments. Y.I. and Y.T. performed single-channel analyses, with support from H.I., and M.I., Y.I., Y.T., S.I.,

M.O., and I.S. designed experiments, analyzed data, and wrote the manuscript with input from all other authors.

## Competing interests

The authors declare no competing interests.
