## [Peer Review File · Nature Communications]

REVIEWER COMMENTS

Reviewer #1 (Remarks to the Author):

In this manuscript, Iwahashi et al. characterize the structural and functional changes introduced by mutations along the two transmembrane helices of KcsA. These mutations are selected based upon sequence alignment to known disease relevant sites in Kv1.1 and shaker voltage-gated potassium channels. KcsA is a common model system to study Shaker channel behavior as it undergoes a similar allosteric regulation and C-type inactivation mechanism. KcsA has the benefit of being pH activated, allowing the authors to study changes in the regulatory mechanism under conditions reproducible (though not perfectly) in an NMR tube. They first characterize the A111V mutant, located along TM2 of the WT protein using liquid-state NMR. They confirm that spectra, and thus structure of the A111V mutant at pH 3 is similar to the WT protein at pH 6.5. This confirms the same locations known to introduce long-QT syndrome in mammals are also relevant to KcsA. They then proceed to interrogate other mutants along TM2 including V91I and V106I. They identify key contacts between these residues and residues along TM1. This is important as the TM1 and TM2 helices of KcsA line up in a similar manner to the animal Kv channels. They further mutate some of these TM1 sites, including A28V, A32V, and T33I. They then seek to tie the states observed in NMR to electrophysiological recordings. Overall this is a well written manuscript and makes a convincing case from the NMR standpoint. To my knowledge, this is the first report of using NMR to probe KcsA mutations away from the selectivity filter. To date, most investigations focused upon selectivity filter, "bow string", or pore helix sites in the protein. Here they interrogate a regulatory hotspot previously untouched by structural techniques like NMR. I recommend its publication after minor revisions.

I suggest the authors address the following points:

1) The authors state in the last paragraph of the Introduction: "...by solution nuclear magnetic resonance (NMR) spectroscopy, which can characterize the conformational dynamics of K⁺ channels in a physiological solution environment."

However, channels are physiologically found in lipid bilayers. The authors do not state how KcsA is solubilized. I assume they are using detergents as in past work; however, this should be explicitly mentioned. KcsA is known to have bound anionic nonannular lipids, thus the bilayer mimetic should be considered. It may be more appropriate to state they are studying KcsA in a physiologically relevant buffered solution in detergent micelles? Or if these are lipid bicelles that should also be stated.

2) I agree that the spectra of A111V at pH 3 is much closer to the WT at pH 6.5 compared to WT at pH 3.0. However, there are some peaks missing at low pH (at least as presented). Past work found that many peaks will disappear in low pH spectra of the channel. Many of those peaks are present in the C-terminal 4 helical bundle, which should pucker out and becomes more dynamic. Can the authors directly account for a few of these resonances in the spectra? While changes in the C-terminus of KcsA do not directly inform our understanding of Kv channels, this information could help us to understand gating motions of many other channels with cytoplasmic domains.

3) Page 6, 2nd paragraph, 2nd sentence: "Referring to the previous results that the gating property is highly sensitive to the volume of the side chain at this position", is awkwardly written, consider revising.

4) On page 8: if the L46I and V91I mutant are showing signal split for Leu59 δ 1 and Val76 γ 1 indicating some conformational change between C and P states, it is possible that channel could have some basal activity (though significantly less than WT). The authors do not provide the single-channel current trace for the V91I mutant, but they did not have the result for L46I.

4) The electrophysiological traces appear to have several different levels of conductance. During the full burst period, there appears that there is not a single precise current level, but, perhaps as many as ~3 different levels within the conductive burst periods (i.e. figure 3B V106I shows several

states that are not fully conductive nor non-conductive). Contrasted to the non-conductive state, where the signal returns to a baseline level each time. Could this mean that there are several conductive conformations of KcsA in these assays that NMR is unable to detect? Perhaps it is possible to have a histogram fit with gaussians to quantify these levels. The treatment of KirBac1.1 (of which the corresponding author is an expert) as presented in Cheng et al. is a possible example. (J Gen Physiol. 2009 Mar; 133(3): 295–305. doi: 10.1085/jgp.200810125 PMID: PMC2654083 PMID: 19204189).

5) It would be informative to provide the full spectra of all mutants studied in the supporting information.

6) On page 9 line 3 from bottom: "To further support this concept ..." they state that A32 and T33 were mutated to bulkier amino acids. They do not state the exact mutations until the final paragraph on page 10. It might improve the readability if the exact mutants are stated on page 9.

7) While the authors prove their point, I am curious how moving to even bulkier sidechains may impact channel function. Were any aromatic amino acids considered?

Reviewer #2 (Remarks to the Author):

Description and summary:

The focus of this MS is on the KcsA potassium channel, a well-established model for structure and function of K-channels spanning all biological kingdoms. Due to the importance of K-channels in cellular homeostasis, structure-function relations of this important class of membrane proteins are of significant importance and have impact on understanding disease involving these channels. KcsA and other K-channels have been studied by a variety of structural and biophysical methods; here the authors employ nuclear magnetic resonance (NMR) to illuminate delicate aspects of channel function. NMR has been used in the past by several groups, including the authors of the MS under scrutiny, with nice results, although the size and complexity of the channel assembly and the need to solubilize it in a membrane-mimicking environment is a constant worry and confounding factor. Thus, the work presented by the authors deserves recognition for importance, timeliness and challenge.

The authors set out to prove, using an extensive series of single-site KcsA mutants, the effects of sterically-demanding residues in key positions in the channel upon the stability of the various channel states. To briefly recapitulate, the channel is known to adopt three well-documented states, closed, permeable and impermeable. As has been shown previously, the ^{13}C - ^1H -HMQC spectrum of a methyl protonated, otherwise deuterated channel is perhaps practically the only NMR experiment suitable for the ultra-large channel assembly. Each of the states has a unique arrangement of cross-peaks in this spectrum that serve as a 'fingerprint'. By introducing the desired sterically-demanding mutations, the authors monitor the effect of each mutation on the NMR spectrum to determine how the population distribution between the states is shifted. This approach has been used in the past; what is novel here is the large number of mutants involved in the study (close to 30) and the notion of systematically examining a particular aspect of structure – that of steric hinderance upon channel function.

The disease-related mutation A111V serves as a proof-of-concept, as it clearly shifts the channel to the closed state and this is easily observed using the HMQC spectrum. Then, a series of V-to-I or L-to-I mutations lead to the identification of four positions, two close to the extracellular side and two mid-TM, for which the channel state is affected. The effect of temperature on these shifts is examined as well. The two mid-TM mutants form VDW contacts with additional residues; in a handsome confirmation of their findings, the authors show that other mutations affecting this region will result in the same effect on channel state.

The authors proceed to provide electrophysiological single-channel measurements of KcsA in wildtype form and three mutant channels. The conductive/non-conductive ratio is shifted from ~5:1 to somewhere between 1:1 and 0.1:1, a dramatic change. A nice correlation

($R^2=0.88$) is seen between the NMR peak intensities (indicating populations) and the electrophysiology results, strengthening their validity. This is built in to a general model for channel behavior (not derived from the current study), in which the channel can be either in the non-conductive state between bursts, with slow kinetics to the burst state, or in a faster equilibrium between conductive and non-conductive states, associated with the permeable and closed states involving changes only in the intracellular gate and not the selectivity filter. In this model, the bulky sidechains appear to affect the population of closed and permeable states within the burst phase, but no clear pattern of effect is demonstrated on the intra-burst state. (See concern below.)

General evaluation:

Investigations of membrane-proteins by NMR are always challenging, and this is true even for the well-behaved KcsA. The work presented here is extensive, encompassing ~30 single-site mutants and background experiments for stereospecific assignment of the methyl probes. The combination of these structural studies and the channel readings provides a comprehensive structure-function view of the channel, and focuses on an aspect that has not been previously addressed (to the best of my knowledge). In summary, this is an impressive feat of research and should be considered for publication. I would like, however, to mention three concerns arising when reading the MS:

Point 1: The culmination of the paper is the model of channel activity described above, and it is presented as a highlight and main advance of the MS. In looking through the references mentioned in this section of the text it is difficult to understand where exactly the existing knowledge ends and the study's novel findings begin. Furthermore, this reviewer fails to see the connection between this new model and the body of data accumulated in the study. Is there a correlation between the various mutations and the model, or is their affect limited to the ratio between closed and permeable states? Or perhaps this model is a separate finding unrelated to the earlier part of the MS? This should be presented more clearly, and the connection between the earlier findings and the model better explained.

Point 2: The authors quote a value of 160 s⁻¹ for the closed-to-permeable exchange equilibrium, and state that this is consistent with the slow-exchange appearance of the spectrum. Eyeballing the few peaks in Figure 2, the difference in chemical shifts are ~0.5 ppm in the ¹³C dimension, or 60-100 Hz depending on the magnetic field (should be stated in the legend but does not appear there), Some exchange broadening would be expected in this situation. This point should be clarified and addressed. For example, differences in the spectra observed at 500/800 MHz could clearly establish the exchange regime and confirm the quoted exchange rate obtained from single-channel currents.

Point 3: Although the authors do not emphasize this, the channel studied here is solubilized in dodecyl maltoside (DM) micelles. This has been the system of choice for the authors in their previous publications on the KcsA channel. Finding this information was not easy – it is omitted from the Materials & Methods (where it should appear explicitly or in reference to a previous paper) and is only mentioned once in the context of the A111V mutant. DM is a mild detergent with a low CMC and forms micelles that are a convenient membrane-mimicking system for NMR studies. Still, as the authors state themselves (p. 14, Discussion 2nd paragraph) the activity of the K-channel is influenced by the membrane environment (lipids), and the DM micelle differs from the natural phospholipidic membrane in width and in composition. True, the correlation obtained with single-channel readings gives credence to the results in the DM micelle, but it is plausible that a different choice of surrogate membrane (e.g. lipoprotein nanodiscs) would change the results in the series of sterically-demanding mutations. It might be difficult to draw subtle conclusions regarding the interplay of gates and the influence of transmembrane steric effects in a setting which is less-than-native. It may not be possible to examine this experimentally within the scope of this paper, but this point should be addressed somehow in the text and the membrane-mimicking system chosen stated more clearly.

In summary – this is a structure-function study of the KcsA channel with an impressive work content, novelty and importance. Before publication, the paper must be further improved by addressing these questions.

Reviewer #3 (Remarks to the Author):

In this manuscript, Yuta Iwahashi et al. studied a hotspot in potassium channel; researchers have previously discovered the association between the mutations at this position and human diseases. The authors studied this hotspot in a model potassium channel KcsA from bacteria and found out that the A111V mutant cannot enter the activated (P) state at a low pH in contrast to WT. They further demonstrated that the bulky side chain of the V111 prevents the channel from shifting from the closed-conductive (C) state to the P state, due to steric contact between residues in the transmembrane helix. Therefore, they concluded that the closure at this activation gate (HBC gate) — rather than the inactivated state of the channel — causes the non-conducting behavior in the A111V mutant. They hypothesized that a similar mechanism might explain the behavior of mutants in Shaker and Kir1.1 channels: the change of the side chain in various mutants could affect the equilibrium between the P state and C state and thus the conducting behavior. They also demonstrated that the exchange between the P state and C state causes the intra-burst gating transition observed in the single channel electrophysiology experiments. I find these findings very interesting as they provide a different perspective on the malfunction of potassium channel and single channel electrophysiology. These findings are well supported by their data. I have several questions and some minor suggestions below:

Q1: In this study, Leu59 and V76 peaks were mainly used as fingerprints for indicating the various states of the protein. Even though these residues are coupled to proton binding and are sensitive to both potassium and proton, they are quite distant from the proton binding sites. Have you studied the proton sensor, H25, to see if it gets protonated at a low pH in the A111V mutant, as your group did for WT before? Will the pKa of H25 change in A111V? Even though it may not be necessary, I would recommend more experiments to ensure that HBC is in the closed conformation in A111V at pH3.5?

Q2: In the WT-KcsA, the interactions between charged residues, such as H25, E118, E120, and R121, have been hypothesized to stabilize the closed HBC. As shown in Figure 1B, Leu24 and V126 show chemical shift change due to the low pH in the A111V mutant. It seems to suggest that the salt bridge and hydrogen bonding network of the charged residues are impaired and there is a conformational change at the pH sensor region. Would that mean the HBC gate is probably partially open, but is not large enough for K⁺ to pass through?

Q3: In Figure 2C panel for V106I at 45 °C, the chemical shift for V76 is in between the values for the P state and C state. As the temperature drops, the peak split into two peaks corresponding to the P state and C state. Does it mean that V76 is in a fast exchange between the P state and C state at 45 °C? But how would the P and C states exchange directly based on the proposed model?

Q4: How many single channel recording current traces were used for the analysis? It is worth noting that different modal conducting behaviors were observed in WT-KcsA (Chakrapani S, Cordero-Morales JF, Perozo E. A quantitative description of KcsA gating II: single-channel currents. *J Gen Physiol.* 2007;130(5):479-496. doi:10.1085/jgp.200709844)

Q5: The conducting and non-conducting dwell times in the intra-burst are on the millisecond time scale. Would you expect to see line broadening of the NMR peaks due to this millisecond exchange process?

Even though it is known that the activation of the HBC gate is quite fast (millisecond time scale), do you think the transition between the closed and open conformations of HBC is also fast?

Minor comments

1. Since Leu59 and Val76 were mainly investigated in the paper, it is probably better to also highlight their positions in the main figure such as in the crystal structure in Figure 1.
2. The conformation of KcsA is dependent on potassium concentration. Were all the experiments done at 100 mM K⁺? I think you need to mention the [K⁺] where you discussed the sample conditions in the main text, at least at the very beginning.
3. Is the protein in its full-length or truncated form? It would be nice to clarify this.
4. In the figure caption of Figure 1, "L25" is a typo. It should be "L24".

Point-by-point responses to reviewers

We would like to thank all of the reviewers for their positive comments and suggestions for our manuscript. According to the reviewers' comments, we have revised the manuscript carefully. Our point-by-point responses are summarized below.

Reviewer #1 (Remarks to the Author):

In this manuscript, Iwahashi et al. characterize the structural and functional changes introduced by mutations along the two transmembrane helices of KcsA. These mutations are selected based upon sequence alignment to known disease relevant sites in Kv1.1 and shaker voltage-gated potassium channels. KcsA is a common model system to study Shaker channel behavior as it undergoes a similar allosteric regulation and C-type inactivation mechanism. KcsA has the benefit of being pH activated, allowing the authors to study changes in the regulatory mechanism under conditions reproducible (though not perfectly) in an NMR tube. They first characterize the A111V mutant, located along TM2 of the WT protein using liquid-state NMR. They confirm that spectra, and thus structure of the A111V mutant at pH 3 is similar to the WT protein at pH 6.5. This confirms the same locations known to introduce long-QT syndrome in mammals are also relevant to KcsA. They then proceed to interrogate other mutants along TM2 including V91I and V106I. They identify key contacts between these residues and restudies along TM1. This is important as the TM1 and TM2 helices of KcsA line up in a similar manner to the animal Kv channels. They further mutate some of these TM1 sites, including A28V, A32V, and T33I. They then seek to tie the states observed in NMR to electrophysiological recordings. Overall this is a well written manuscript and makes a convincing case from the NMR standpoint. To my knowledge, this is the first report of using NMR to probe KcsA mutations away from the selectivity filter. To date, most investigations focused upon selectivity filter, "bow string", or pore helix sites in the protein. Here they interrogate a regulatory hotspot previously untouched by structural techniques like NMR. I recommend its publication after minor revisions. I suggest the authors address the following points:

1) The authors state in the last paragraph of the Introduction: "...by solution nuclear magnetic resonance (NMR) spectroscopy, which can characterize the conformational dynamics of K⁺ channels in a physiological solution environment."

However, channels are physiologically found in lipid bilayers. The authors do not state how KcsA is solubilized. I assume they are using detergents as in past work; however, this should be explicitly mentioned. KcsA is known to have bound anionic nonannular lipids, thus the bilayer mimetic should be considered. It may be more appropriate to state they are studying KcsA in a physiologically relevant buffered solution in detergent micelles? Or if these are lipid bicelles that should also be

stated.

We appreciate the reviewer's comment. We added the sample details in the main text and the Materials and Methods section to clarify that we analyzed KcsA in detergent micelles. We also changed the last paragraph of the introduction according to the reviewer's suggestion, as follows.

Introduction (p. 4, line 13- p.5, line 2)

“Here, we investigated the structural mechanism underlying the altered gating behavior of K⁺ channels induced by the hot spot mutation, by solution nuclear magnetic resonance (NMR) spectroscopy, which can characterize the conformational dynamics of K⁺ channels **in a physiologically relevant solution environment.**”

Methods (p.21, line 21- p.22, line 7)

The KcsA protein was solubilized in 20 mM DDM (Dojindo) and purified to homogeneity by chromatography on HIS-Select Nickel Affinity Gel (Sigma-Aldrich). The His-tags were with HRV-3C protease (Novagen), and the cleaved His-tags and the protease were removed on HIS-Select Nickel Affinity Gel. NMR samples were prepared by changing the buffer to 20 mM potassium phosphate, 100 mM KCl, by sequential dilution and concentration with an Amicon[®] Ultra Centrifuge Filter Unit, NMWL 50 kDa (Merck Millipore). All NMR spectra were recorded in buffer containing 20 mM potassium phosphate (pH 3.0 or 6.5), 100 mM KCl, and ~ 5 mM DDM in 100% D₂O. The pH values in D₂O were calibrated by adding 0.4 pH unit to the reading on the pH meter⁵².

2) I agree that the spectra of A111V at pH 3 is much closer to the WT at pH 6.5 compared to WT at pH 3.0. However, there are some peaks missing at low pH (at least as presented). Past work found that many peaks will disappear in low pH spectra of the channel. Many of those peaks are present in the C-terminal 4 helical bundle, which should out and becomes more dynamic. Can the authors directly account for a few of these resonances in the spectra? While changes in the C-terminus of KcsA do not directly inform our understanding of Kv channels, this information could help us to understand gating motions of many other channels with cytoplasmic domains.

We appreciate the reviewer's careful reading of our manuscript. As the reviewer pointed out, the methyl signals from the C-terminal helical bundle region (Val126, Leu151, and Leu155) disappeared in the A111V mutant at pH 3.0 and 45 °C, and new signals were observed at the chemical shifts of the wild type at pH 3.0 (Supporting figure 1). For

example, we clearly observed the disappearance of the Val126 γ 1 and γ 2 signals in the A111V mutant at pH 3.0, and the appearance of two new signals at the chemical shifts of the Val126 γ 1 and γ 2 of the wild type at pH 3.0 (Supporting figure 1, red stars). Although it was hard to trace the Leu151 and Leu155 methyl signals due to the severe spectral overlap, these methyl signals were likely to be observed at the chemical shifts of those of the wild type at pH 3.0. These results strongly suggest that the pH-dependent structural changes occur at the C-terminal helical bundle region in the A111V mutant at pH 3.0, as also observed in the wild type, presumably reflecting the local structural changes induced by the protonation of the protonatable residues, such as H124, H128, and H145 in the C-terminal region.

Supporting figure 1 Enlarged view of the Leu/Val region of the ^1H - ^{13}C HMQC spectra of the wild type and the A111V mutant. (A) Enlarged view of the Leu/Val region of overlaid ^1H - ^{13}C HMQC spectra of the wild type at pH 6.5 (blue) and the A111V mutant at pH 3.0 (black). The spectra were measured at 45 °C and 18.8 Tesla (800 MHz ^1H frequency), in the presence of 100 mM KCl. Representative residues that were observed in the well-resolved region are labeled. The chemical shifts of Leu24, Val126, Leu151, and Leu155 of the wild type at pH 3.0 are indicated as red stars. (B) Mapping of the highlighted residues on the crystal structure of the full-length KcsA in the closed state (PDB ID: 3EFF).

Our previous pH titration experiments demonstrated that the pH-dependent structural changes in the C-terminal region are not coupled to the structural changes at the transmembrane pore region, as the $\text{pH}_{1/2}$ value of the signals from the transmembrane pore (5.0 ± 0.3) was different from that of the C-terminal region (6.4 ± 0.4) (Supporting figure 2) (Imai *et al.*, *PNAS* 107(14), 6216–6221 (2010)). Our NMR results indicated that these pH-dependent C-terminal structural changes actually occurred in the A111V mutant, while those in the transmembrane pore were significantly perturbed. It should be noted that, in the electrophysiological analyses of KcsA (Chakrapani *et al.*, *J. Gen. Physiol.* 130(5), 465–478 (2007), Chakrapani *et al.*, *J. Gen. Physiol.* 130(5), 479–496 (2007)), the $\text{pH}_{1/2}$ for the activation was 4.2–4.5, which matched the $\text{pH}_{1/2}$ of the transmembrane pore. These findings suggest that the pH-dependent structural changes at the C-terminal helical bundle region are not critical for the pH-dependent gating of KcsA.

[REDACTED]

To clarify this point, we added a brief introduction for the C-terminal region of KcsA, an

explanation for the pH-dependent spectral changes in the A111V mutant, and a supporting figure.

Results (p.5, line 15 - p.6, line 2)

In this study, we used the pH-dependent K⁺ channel from *Streptomyces lividans*, KcsA, to investigate the gating mechanisms of the Kv channels²⁵. KcsA shows similar electrophysiological properties to those observed in human Kv channels, and thus serves as a prototypical counterpart of eukaryotic Kv channels^{26,27}. Moreover, KcsA lacks a voltage-sensor domain and is composed solely of the transmembrane pore region, highlighting its suitability for investigating the effects of the hot spot mutation on the structure and dynamics of the pore region. KcsA is a homo-tetrameric channel, and each subunit is composed of three transmembrane helices, referred to as the outer helix, pore helix, and inner helix, forming the transmembrane K⁺-pore, followed by the C-terminal intracellular region that stabilizes the tetrameric structure and plays a modulatory role in the pH-dependent gating^{8,28-30}.

Results (p.7, line 16 - p.8 line 9)

Interestingly, some methyl signals did not follow this trend. These exceptional methyl groups are located at the cytoplasmic end of the outer transmembrane helix (Leu24) and in the C-terminal helix bundle region (Val126, Leu151, and Leu155), and exhibited the same or similar chemical shifts as those of the wild type at pH 3.0 (red stars, Supplementary Fig. 3). In our previous pH titration analyses of KcsA in DDM micelles, we demonstrated that the C-terminal region exhibited a pH-dependent structural transition ($\text{pH}_{1/2} = 6.4 \pm 0.4$) that was not coupled to the pH-dependent gating transition at the transmembrane pore ($\text{pH}_{1/2} = 5.0 \pm 0.3$)¹⁰. In addition, Leu24 is located next to the pH sensor residue, His25, and is expected to be sensitive to the perturbation of the salt-bridge network formed by His25 under acidic conditions^{31,33}. These results indicate that the A111V mutant is likely to be protonated at pH 3.0, which caused the pH-dependent structural transitions at the C-terminal region and the perturbation of the salt-bridge network centered on His25, but could not induce the structural changes of the transmembrane pore, and thus the pore region is forced to adopt the C state conformation, regardless of the pH condition. As the C state represents the non-conductive structure, in which the K⁺-permeation is blocked by the constricted HBC gate, our results indicate that the A111V mutant is a non-conductive mutant whose HBC gate adopts the closed structure even under acidic conditions.

Supplementary information (p. 4)

Supplementary Fig. 3 Enlarged view of the Leu/Val region of the ^1H - ^{13}C HMQC spectra of the wild type and the A111V mutant. (A) Enlarged view of the Leu/Val region of overlaid ^1H - ^{13}C HMQC spectra of the wild type at pH 6.5 (blue) and the A111V mutant at pH 3.0 (black). The spectra were measured at 45 °C and 18.8 Tesla (800 MHz ^1H frequency), in the presence of 100 mM KCl. Representative residues observed in the well-resolved region are labeled. The chemical shifts of Leu24, Val126, Leu151, and Leu155 of the wild type at pH 3.0 are indicated as red stars. (B) Mapping of the highlighted residues on the crystal structure of the full-length KcsA in the closed state (PDB ID: 3EFF)⁶¹.

3) Page 6, 2nd paragraph, 2nd sentence: “Referring to the previous results that the gating property is highly sensitive to the volume of the side chain at this position”, is awkwardly written, consider revising.

We appreciate the reviewer’s suggestion. We rewrote the sentence as follows.

Results (p.6, lines 21-24)

The sequence alignment of the KcsA and Kv channels revealed that the hot spot residue in KcsA is Ala111, which corresponds to Val408 in Kv1.1 and Val478 in Shaker Kv (Figure 1A). Kitaguchi *et al.* reported that the gating behavior in Shaker Kv channels was highly

variable, depending on the volume of the side-chain at this position¹⁵, and thus we replaced Ala at position 111 with Val and characterized the structure of the mutant KcsA.

4) On page 8: if the L46I and V91I mutant are showing signal split for Leu59 δ I and Val76 γ I indicating some conformational change between C and P states, it is possible that channel could have some basal activity (though significantly less than WT). The authors do not provide the single-channel current trace for the V91I mutant, but they did not have the result for L46I.

We appreciate the reviewer's comment. Yes, the V91I mutant does have basal activity that is lower than that of the wild type. The single-channel current of the V91I mutant is shown in Figures 4A and 4B. Although its net open probability during the burst period was significantly lower than that of the wild type, the V91I mutant could adopt the K⁺-conductive state and permeate K⁺, corresponding to the "basal activity" as the reviewer pointed out.

Although our attempt to record the single-channel current of the L46I mutant was not successful, due to the instability of the patch, the formation of the permeable state in the L46I mutant strongly suggests that it exhibits similar basal activity to that observed in the V91I mutant. Furthermore, it should be noted that the side-chain of Leu46 forms a van der Waals contact with that of Val91, in the crystal structure of KcsA in the closed conformation (Supporting figure 3). Thus, we believe that this common mechanism can explain the shift in the equilibrium between the P and C states, observed in the V91I and L46I mutants.

Supporting figure 3 Side-chain interaction formed between L46 and V91 in the crystal structure of the closed conformation (PDB ID: 1K4C). The L36, V106, and A111 residues are also displayed.

To clarify this point, we added an explanation for the basal activity, as follows.

Results (p.13 lines 2-5)

Notably, as compared to the wild type, the V91I mutant exhibited significantly lower basal activity, which could be characterized by the frequent transitions between the conductive and non-conductive states and the increased population of the non-conductive state during the burst period (Figure 2A, blue dotted line).

4) The electrophysiological traces appear to have several different levels of conductance. During the full burst period, there appears that there is not a single precise current level, but, perhaps as many as ~3 different levels within the conductive burst periods (i.e. figure 3B V106I shows several states that are not fully conductive nor non-conductive). Contrasted to the non-conductive state, where the signal returns to a baseline level each time. Could this mean that there are several conductive conformations of KcsA in these assays that NMR is unable to detect? Perhaps it is possible to have a histogram fit with gaussians to quantify these levels. The treatment of KirBac1.1 (of which the corresponding author is an expert) as presented in Cheng et al. is a possible example. (J Gen Physiol. 2009 Mar; 133(3): 295–305. doi: 10.1085/jgp.200810125 PMID: 19204189).

We appreciate the reviewer's suggestion. We reanalyzed the single-channel data according to the reviewer's suggestion and added all point histograms in Figure 3B.

As the reviewer pointed out, we observed multiple conductance levels in the single-channel trace of the V106I mutant. We were not able to clearly define the number of different conductance levels that the V106I mutant could adopt from the all point histogram, because of the limited quality of our electrophysiological recording.

Interestingly, the broader distribution in the K⁺-conductive state was similarly observed in the V91I mutant. Based on these observations, we presume the broad distribution in the conductance level is a functional feature of the KcsA mutants, in which the closed state is more populated than that in the wild type.

These multiple conductance levels may be attributable to the conformational heterogeneity of the helix bundle crossing (HBC) structure in the mutants, because the HBC can adopt various asymmetric structures, depending on the number of subunits adopting the open conformation among the 4 subunits in the tetramer, assuming that the 4 subunits in the tetramer are not fully cooperative in the open-closed structural transition. This

conformational heterogeneity could be one of the reasons for the disappearance of the methyl signals corresponding to the HBC region in the mutants, as stated below (please refer to our response to Q.1 raised by reviewer 3). These kinds of asymmetric behaviors were previously characterized in KcsA and KcV channels, by using a hetero-tetrameric protein containing different combinations of wild-type and mutant subunits (Rotem *et al.*, *J. Gen. Physiol.* 135(1), 29–42 (2009), Tan *et al.*, *FEBS Lett.*, 584(8), 1602–1608 (2010)), and different conductance levels were actually observed between the hetero-tetrameric proteins with different compositions. Although these analyses focused on the asymmetry in the selectivity filter structure, we expect that the asymmetry in the HBC structure could also cause multiple conductance levels.

In the revised manuscript, we added the all points histograms in Fig. 3B. We reanalyzed the populations of the conductive and non-conductive states based on the two-state Gaussian fitting of the distributions, rather than a classification using a single threshold value, and revised Fig. 3C using the newly obtained conductive/non-conductive populations. We also described the possible source of the multi-conductance levels in the Discussion section.

Result (p.13, lines 6-16)

The K⁺-current exhibited a broader distribution than in the wild type as shown in the all-points histogram, suggesting there are multiple sub-conductance levels in the V91I mutant. From the fitting of the distribution of the V91I mutant assuming two states, conductive and nonconductive states, the populations of these states were calculated to be 30 % and 70 %, respectively (Fig. 4A bottom, red and blue dotted lines), indicating that the relative populations of these two states during the burst period are inverted in the V91I mutant as compared to the wild type. Considering the NMR results that, in the V91I mutant, the population of the C state markedly increased whereas the population of the I state was not largely affected, these results suggest that the intra-burst gating transitions are attributable to the exchange between the P and C states, rather than that between the P and I states (Figs. 2C and 4A).

Result (p.13, lines 22-24)

The broad distribution of the K⁺-conductive state was similarly observed in the V106I mutant, showing the existence of the multiple sub-conductance levels as observed in the V91I mutant.

Discussion (p.20, lines 7-19)

Finally, in the single-channel analyses of the V91I and V106I mutants, we observed multiple sub-conductance levels during the burst gating (Fig. 4B). These multiple conductance levels may be attributable to the conformational heterogeneity of the HBC structure in the mutants, because the overall HBC gate can form multiple asymmetric structures, depending on the number of subunits adopting the open conformation among the 4 subunits in the tetramer, assuming that the 4 subunits in the tetramer are not fully cooperative in the open-closed structural transition. The functional consequences of the asymmetric structures were characterized for the KcsA and KcV channels, by using a hetero-tetrameric protein containing different combinations of wild-type and mutant subunits^{55,56}, and various conductance levels were observed between the hetero-tetrameric proteins with different compositions. Although these analyses focused on the asymmetry of the SF gate, the asymmetry of the HBC gate could also cause multiple sub-conductance levels.

Figures (p.35)

Fig. 4 Single-channel analyses of KcsA. (A) Representative single-channel current traces of the wild type and the V91I mutant of KcsA, obtained with the planar lipid bilayer system. Currents were recorded at -200 mV with a symmetric K^+ concentration of 100 mM. The asymmetric pH conditions (pH 6.5/3.0) were used to observe the K^+ -currents of KcsA, with the intracellular side directed toward the acidic side. The burst period and the inter-burst nonconductive state are denoted as gray and green half-parentheses, and the conductive and non-conductive states during the burst period are marked with the blue and red dotted lines, respectively. (B) Open probabilities during the burst period in the wild type and the V91I, V106I, and A111V mutants. All points histograms of the channel openings from each recording are shown below. The open probabilities were calculated from the deconvolution assuming two Gaussian distributions corresponding to the conductive and non-conductive states (dotted lines). (C) Correlation plot between the fraction of the P state in the NMR

analyses and the fraction of the conductive state during the burst 1.iod in the single-channel analyses ($R^2 = 0.89$). The error bars represent one standard deviation, as estimated from a covariance matrix. The fractions of the P state are calculated from the signal intensities of the P state divided by the sum of the intensities of the P and C states, using the NMR spectra recorded at 25 °C and pH 3.0, in the presence of 100 mM K⁺. (D) Schematic model of the single-channel gating behavior described by the three different conformational states of KcsA. Source data are provided as a Source Data File.

5) It would be informative to provide the full spectra of all mutants studied in the supporting information.

We appreciate the reviewer for this suggestion. We added a supporting figure showing the full spectra of all mutants (A28V, A32V, T33I, L36I, L46I, V91I, V106I, and A111V).

Supplementary information (p.3)

Supplementary Fig. 2 NMR spectra of KcsA mutants. ^1H - ^{13}C HMQC spectra of KcsA mutants (A28V, A32V, T33I, L36I, L46I, V91I, V106I, and A111V) are shown. An overlay of the ^1H - ^{13}C HMQC spectra of the wild type KcsA measured under different conditions (blue: pH 6.5 and 45 °C, red: pH 3.0 and 45 °C, in the presence of 100 mM KCl) is also shown. The Ile δ 1, Leu δ 1/2, and Val γ 1/2 methyl groups were selectively labeled with ^1H and ^{13}C in the wild type and the A28V, A32V, T33I, and A111V mutants, and the Leu δ 1/2 and Val γ 1/2 methyl groups were labeled in the L36I, L46I, V91I, and V106I mutants, in an otherwise highly deuterated background. The spectra were recorded at pH 3.0 and 45 °C in the presence of 100 mM KCl, at 11.7 Tesla (500 MHz ^1H frequency for L36I), 14.1 Tesla (600 MHz ^1H frequency, for A28V, A32V, and L46I), or 18.8 Tesla (800 MHz ^1H frequency, for wild type, T33I, V91I, V106I, and A111V).

6) On page 9 line 3 from bottom: “To further support this concept ...” they state that A32 and T33 were mutated to bulkier amino acids. They do not state the exact mutations until the final paragraph on page 10. It might improve the readability if the exact mutants are stated on page 9.

We appreciate the reviewer’s suggestion. We added the names of the exact mutations we used, as follows.

Results (p. 11, lines 9-11)

To further support this concept, we designed **two additional mutants, A32V and T33I, in which Ala32 and Thr33 were mutated to a bulkier valine or isoleucine residue.**

Results (p. 11, lines 17-20)

We also mutated Ala28 **to valine**, as its side-chain is farther from the mid-transmembrane cluster in the HBC-open structure, to serve as a negative control (7.8 Å between Ala28C β and Leu105C β in the open structure, as compared to 6.1 Å between Ala28C β and Leu105C β in the closed structure) (Fig. 3B).

7) While the authors prove their point, I am curious how moving to even bulkier sidechains may impact channel function. Were any aromatic amino acids considered?

We mainly utilized a mutation to a methyl-containing side-chain in this study, rather than a mutation to a bulkier aromatic side-chain. The reasons for this are as follows: (i) the mutation to an aromatic residue can induce large ring-current ^1H shifts in nearby methyl-probes, which can complicate the interpretation of the NMR spectrum of KcsA. We

wanted to minimize the direct influence of the mutation on the methyl chemical shift, as the methyl ^1H chemical shift distribution of KcsA was inferior due to the alpha-helical structure, and only a few methyl probes were useful as a fingerprint. (ii) The site-directed mutagenesis of methyl-containing side-chains is reportedly useful to investigate an allosteric network at various locations throughout the protein, because methyl-containing amino acids are widely distributed throughout the sequence and their mutations can mildly perturb the allosteric network (Aoto *et al.*, *Sci. Rep.* 6:28655 (2016), Shi and Kay *PNAS* 111(6), 2140–2145 (2014)). This strategy was successfully applied to investigate the allosteric communications in p38 γ kinase and HsIV protease. Although we have not analyzed mutations to bulkier aromatic amino acids in KcsA, the V478W mutation (V478 of Shaker corresponds to A111 of KcsA) reportedly caused lower K^+ -permeability in the Shaker channel (Kitaguchi *et al.*, *J. Gen. Physiol.* 124, 319–332 (2004)), and thus we expect that a similar trend would be observed in KcsA (Supporting figure 4).

[REDACTED]

Regarding this point, we briefly explained the reason why we utilized methyl mutations as follows.

Results (p.9, lines 5-9)

To further investigate the mechanism by which the bulkiness of the side-chain in the transmembrane region affects the gating of KcsA, we systematically mutated the methyl-containing residues in the transmembrane region to those with different side-chain

volumes, as mutations of the methyl-containing residues were proposed to be useful for investigating allosteric communication at various locations throughout the protein^{34,35}.

Reviewer #2 (Remarks to the Author):

Description and summary:

The focus of this MS is on the KcsA potassium channel, a well-established model for structure and function of K-channels spanning all biological kingdoms. Due to the importance of K-channels in cellular homeostasis, structure-function relations of this important class of membrane proteins are of significant importance and have impact on understanding disease involving these channels. KcsA and other K-channels have been studied by a variety of structural and biophysical methods; here the authors employ nuclear magnetic resonance (NMR) to illuminate delicate aspects of channel function. NMR has been used in the past by several groups, including the authors of the MS under scrutiny, with nice results, although the size and complexity of the channel assembly and the need to solubilize it in a membrane-mimicking environment is a constant worry and confounding factor. Thus, the work presented by the authors deserves recognition for importance, timeliness and challenge.

The authors set out to prove, using an extensive series of single-site KcsA mutants, the effects of sterically-demanding residues in key positions in the channel upon the stability of the various channel states. To briefly recapitulate, the channel is known to adopt three well-documented states, closed, permeable and impermeable. As has been shown previously, the ¹³C-1H-HMQC spectrum of a methyl protonated, otherwise deuterated channel is perhaps practically the only NMR experiment suitable for the ultra-large channel assembly. Each of the states has a unique arrangement of cross-peaks in this spectrum that serve as a 'fingerprint'. By introducing the desired sterically-demanding mutations, the authors monitor the effect of each mutation on the NMR spectrum to determine how the population distribution between the states is shifted. This approach has been used in the past; what is novel here is the large number of mutants involved in the study (close to 30) and the notion of systematically examining a particular aspect of structure – that of steric hinderance upon channel function.

The disease-related mutation A111V serves as a proof-of-concept, as it clearly shifts the channel to the closed state and this is easily observed using the HMQC spectrum. Then, a series of V-to-I or L-to-I mutations lead to the identification of four positions, two close to the extracellular side and two mid-TM, for which the channel state is affected. The effect of temperature on these shifts is examined as well. The two mid-TM mutants form VDW contacts with additional residues; in a handsome confirmation of their findings, the authors show that other mutations affecting this region will result in the same effect on channel state.

The authors proceed to provide electrophysiological single-channel measurements of KcsA in wildtype form and three mutant channels. The conductive/non-conductive ratio is shifted from ~5:1 to somewhere between 1:1 and 0.1:1, a dramatic change. A nice correlation ($R(\text{squared})=0.88$) is seen between the NMR peak intensities (indicating populations) and the electrophysiology results, strengthening their validity. This is built in to a general model for channel behavior (not derived from the current study), in which the channel can be either in the non-conductive state between bursts, with slow kinetics to the burst state, or in a faster equilibrium between conductive and non-conductive states, associated with the permeable and closed states involving changes only in the intracellular gate and not the selectivity filter. In this model, the bulky sidechains appear to affect the population of closed and permeable states within the burst phase, but no clear pattern of effect is demonstrated on the intra-burst state. (See concern below.)

General evaluation:

Investigations of membrane-proteins by NMR are always challenging, and this is true even for the well-behaved KcsA. The work presented here is extensive, encompassing ~30 single-site mutants and background experiments for stereospecific assignment of the methyl probes. The combination of these structural studies and the channel readings provides a comprehensive structure-function view of the channel, and focuses on an aspect that has not been previously addressed (to the best of my knowledge). In summary, this is an impressive feat of research and should be considered for publication. I would like, however, to mention three concerns arising when reading the MS:

Point 1: The culmination of the paper is the model of channel activity described above, and it is presented as a highlight and main advance of the MS. In looking through the references mentioned in this section of the text it is difficult to understand where exactly the existing knowledge ends and the study's novel findings begin.

We appreciate the reviewer's thoughtful comment.

Previous studies focused on the gating mechanism of KcsA were mainly based on electrophysiological analyses of the KcsA K^+ current or X-ray crystallographic structures. However, as the crystal structure usually represents a static snapshot and cannot capture the dynamic properties of the gate structure, the detailed structural mechanism underlying the multimodal KcsA gating behavior has remained enigmatic. For example, a couple of contradicting reports attributed the fast, intra-burst gating of KcsA to either TM2 structural transitions (Baker *et al.*, *Nat. Struct. Mol. Biol.* 14(11), 1089-1095 (2007)) or SF gate transitions (Chakrapani *et al.*, *Nat. Struct. Mol. Biol.* 18(1), 67-74 (2011)).

In order to address this situation, we utilized solution NMR, which can characterize the

conformational dynamics of proteins, to describe the gating behavior of KcsA from a structural viewpoint. In our previous NMR studies, we mainly focused on a slow-gating transition, on the order of seconds (inter-burst gating), and revealed that this slow gating is attributable to the conformational exchange process of the SF-gate (Imai *et al.*, *PNAS* 107 (14), 6216–6221 (2010), Imai *et al.*, *J. Biol. Chem.* 287(2), 39634-39641 (2012)). However, in these studies, the structural mechanism of the fast, intra-burst gating was not well characterized.

In this manuscript, we first focused on the structural mechanism of the disease-related mutant, A111V, and during the course of our analyses we found a set of novel mutants in which the fast, intra-burst gating is altered. Our NMR and single-channel analyses of these mutants revealed that the intra-burst gating is attributable to the conformational exchange of the HBC gate, and thus we established a revision of the structural model for the single-channel behavior of KcsA.

Furthermore, this reviewer fails to see the connection between this new model and the body of data accumulated in the study. Is there a correlation between the various mutations and the model, or is their affect limited to the ratio between closed and permeable states?

In our NMR and single-channel analyses of the wild type and various mutants, we observed a strong correlation between the P state/C state populations in the NMR analyses and the intra-burst conductive/non-conductive populations in the single-channel analyses; therefore, our model is firmly based on the results obtained with these mutants. As the reviewer pointed out, this correlation was limited to the ratio between the closed and permeable states, and we did not evaluate the correlation between the impermeable and inter-burst long non-conductive states, because it was technically difficult to accurately measure the population of the latter state. However, our model was further validated by the kinetic analysis, in which the exchange rate between the P and C states was consistent with the intra-burst gating kinetics (please refer to our response to point 2). Therefore, we believe that our model is sufficiently supported by the evidence from both the NMR and single-channel analyses.

Or perhaps this model is a separate finding unrelated to the earlier part of the MS? This should be presented more clearly, and the connection between the earlier findings and the model better explained.

As the reviewer pointed out, the Introduction and the first half of the Results are focused on the effects of the disease mutation, but the second half of our manuscript describes the structural model of the single-channel gating behavior of KcsA. However, these two topics are interdependent, because we identified a set of novel mutations during the course of the analyses of the disease-related mutant, A111V, and had to establish a model to explain the altered single-channel behaviors of these mutations.

In order to clarify these points, we have carefully revised the Introduction section, the part of the Results section that describes the structural model for the single-channel gating of KcsA, and the Discussion section, as follows. In the current version, we hope that the readers can refer to the preceding studies and our previous NMR models, and thus clearly identify the novel findings in this study.

Introduction (p. 4, line 23 - p.5, line 11)

Here, we investigated the structural mechanism underlying the altered gating behavior of K⁺ channels induced by the hot spot mutation, by solution nuclear magnetic resonance (NMR) spectroscopy, which can characterize the conformational dynamics of K⁺ channels **in a physiologically relevant solution environment**. Using a prototypical K⁺ channel from *Streptomyces lividans*, KcsA, we demonstrated that the Ala to Val mutation at the hot spot residue greatly stabilizes the structure with the closed HBC gate, by affecting the conformational exchange process accompanying its structural rearrangements.

Furthermore, we identified a set of mutants in which the conformational exchange process is altered to different extents. The NMR characterizations of the exchange process, along with the single-channel analyses of these mutants, allowed us to clarify the structural mechanism of the multi-timescale gating behavior of KcsA. Our results show that the change in the conformational dynamics of the HBC gate underlies the altered gating behavior observed in the disease-related mutants of K⁺ channels.

Results (p.14, line 6 - p.15, line 4)

Although we could observe a positive correlation between the P and C state populations defined by the NMR analyses and the intra-burst conductive/non-conductive populations, it was difficult to quantitatively determine the population of the long, inter-burst non-conductive state due to the insufficient numbers of sampling points of the slow gating transitions, which hampered the analysis of the correlation including this state. Therefore, we compared the exchange kinetics between the P and C states with the intra-burst gating kinetics, to obtain further supportive data. We analyzed the intra-burst gating kinetics of

the V91I mutant and calculated the mean dwell times of the conductive and non-conductive states within the burst period (Supplementary Fig. 4). The mean dwell times of the conductive and non-conductive states were 7.9 and 27.0 ms, respectively, and correspond to an exchange rate of 160 s^{-1} assuming two-state exchange. These kinetic parameters are consistent with the NMR observation that the exchange between the P and C states was in a slow-exchange regime with significant exchange-induced line broadening effects, and cannot be explained by the exchange between the P and I states, as we previously demonstrated that this exchange process is much slower, with a k_{ex} of 1.4 s^{-1} at $40 \text{ }^{\circ}\text{C}$. In the NMR spectrum of the V91I mutant at $25 \text{ }^{\circ}\text{C}$, extra line broadening effects on the order of several tens of seconds were observed for the V76 γ 1 methyl signals in the ^{13}C dimension, as compared to the wild type (Supplementary Fig. 5), consistent with the expected exchange contribution ($= 46.9 \text{ s}^{-1}$) assuming the exchange rate of 160 s^{-1} , the ^{13}C chemical shift difference of 0.45 ppm, and the major and minor state populations of 70 % and 30 % at 18.8 Tesla (800 MHz ^1H frequency)⁴⁰. These results further support our proposal that the exchange between the P and C states explains the intra-burst conductive and non-conductive transitions.

Results (p.15, line6 - p. 16 line 18)

Description of the KcsA single-channel gating behavior by the three distinct conformational states defined by NMR

On the basis of these observations, we modeled the single-channel gating behavior of KcsA under acidic conditions, according to the three distinct conformational states defined by the NMR analyses (Fig. 4D). To date, numerous electrophysiological studies have established that KcsA gating can be modeled by multiple transitions with different time scales, as stated above^{37-39,41}: a slow gating between the burst and the inter-burst non-conductive states on a time scale on the order of seconds, and intra-burst gating on a time scale on the order of milliseconds. However, the detailed structural mechanism of this multi-timescale behavior has remained poorly understood. Regarding the slow gating process, we previously proposed that this gating process could be explained by the slow conformational equilibrium between the P and I states, which differ in the structure of the SF gate, and demonstrated that these two states are exchanging on a time scale on the order of 1 second^{10,30}. This notion is further supported by the results obtained with the E71A and Y82A mutants, in which the P and I states are dominantly populated in our NMR analyses, respectively^{10,30}. Given the electrophysiological results that the E71A mutant does not exhibit the long closure and the Y82A mutant exhibits the prolonged closure interspersed with the shorter burst period^{32,37,39}, we considered that the non-

conductive state between the burst periods (Fig. 4D, green), of which duration time is on the second time scale, corresponds to the I state defined by the NMR analyses.

Regarding the fast, intra-burst gating of KcsA, the structural mechanism is still under debate, and there are several proposed mechanisms in which the gating is described by the conformational transitions of the SF gate or the HBC gate^{20,37,39}. Here, during the course of the characterizations of the disease-related A111V mutant, we identified a set of novel mutants in which the intra-burst gating is altered, and showed that the non-conductive state population within the burst period correlates with the C state population. Furthermore, the kinetics of the exchange between the P state and the C states, estimated from the NMR line-shapes, was also consistent with the intra-burst gating kinetics in the single-channel recordings (Supplementary Fig. 5). These results indicate that the millisecond-order gating transition between the conductive (Fig. 4D, red) and non-conductive states (Fig. 4D, blue) during the burst period corresponds to the exchange between the P and C states defined by the NMR analyses. This model describes the single-channel behavior of KcsA with two different non-conductive conformational states, the I state with the closed SF gate and the C state with the closed HBC gate, and the highly variable single-channel kinetic behavior can be explained by the different lifetimes of these two conformational states.

Point 2: The authors quote a value of 160 s⁻¹ for the closed-to-permeable exchange equilibrium, and state that this is consistent with the slow-exchange appearance of the spectrum. Eyeballing the few peaks in Figure 2, the difference in chemical shifts are ~0.5 ppm in the ¹³C dimension, or 60-100 Hz depending on the magnetic field (should be stated in the legend but does not appear there), Some exchange broadening would be expected in this situation. This point should be clarified and addressed. For example, differences in the spectra observed at 500/800 MHz could clearly establish the exchange regime and confirm the quoted exchange rate obtained from single-channel currents.

We appreciate the reviewer's careful reading of our manuscript.

As the reviewer pointed out, we did observe the line broadening effects in the V76γ1 methyl signals of the V91I and V106I mutants. To clarify this point, we added the 1D projections of V76γ1 in the ¹³C dimension in Supplementary Fig. 5.

It was challenging to quantitatively obtain the exchange rate from these line shapes, because it was difficult to acquire the intrinsic linewidths without the exchange contributions, and conduct quantitative relaxation-based experiments such as ¹³C CPMG relaxation dispersion, due to the lack of sensitivity. However, we were able to roughly

estimate the exchange rate that explained the extent of line broadening observed in the V91I mutant. Here, we focused on the V76 γ 1 methyl signal of the permeable state and the contribution of the exchange between the permeable (major) and closed (minor) states to the exchange-induced line-broadenings, as we previously found that the exchange between the impermeable state was very slow ($k_{\text{ex}} = 1.4 \text{ s}^{-1}$ at 40 °C) (Imai *et al.*, *J. Biol. Chem.* 287(2), 39634-39641 (2012)). From the comparison of the linewidths between the wild type and the V91I mutant, the exchange contributions in the ^{13}C linewidth were estimated to be around 60 s^{-1} , assuming that the exchange contribution in the wild type is minimal because of the small population of the closed state (minor state). If we assumed the ^{13}C chemical shift difference of 0.45 ppm, the minor state population of 30 %, and the exchange rate of 160 s^{-1} , then the exchange contribution in the ^{13}C linewidth at 800 MHz was estimated to be 46.9 s^{-1} , which was agreeably consistent with the experimentally observed line-broadening.

As the reviewer suggested, a comparison of the linewidths between the low and high field data (*i.e.*, 500 MHz and 800 MHz) would be very useful to prove the existence of the exchange contributions on the millisecond timescale. However, as the exchange rate is not as fast as the chemical shift difference in our case ($k_{\text{ex}} \ll \Delta\omega$, where $\Delta\omega$ is the chemical shift difference in rad/s), the magnetic field-dependent change in the spectral linewidth is expected to be minimal. If we assume the same parameters stated above, then the extra line-broadening effect at 800 MHz, as compared to that at 500 MHz, is expected to be $\sim 2 \text{ s}^{-1}$, which would be too small to detect.

We added a supplementary figure and an explanation for the exchange broadenings observed in the V76 γ 1 methyl signal. We also stated the magnetic field strength in the figure legends, as follows.

Results (p.14, line 16 - p.15, line 4)

These kinetic parameters are consistent with the NMR observation that the exchange between the P and C states was in a slow-exchange regime **with significant exchange-induced line broadening effects, and cannot be explained by the exchange between the P and I states, as we previously demonstrated that this exchange process is much slower, with a k_{ex} of 1.4 s^{-1} at 40 °C. In the NMR spectrum of the V91I mutant at 25 °C, extra line broadening effects on the order of several tens of seconds were observed for the V76 γ 1 methyl signals in the ^{13}C dimension, as compared to the wild type (Supplementary Fig. 5), consistent with the expected exchange contribution (= 46.9 s^{-1}) assuming the exchange rate of 160 s^{-1} , the ^{13}C chemical shift difference of 0.45 ppm, and the major and minor state populations of 70 % and 30 % at 18.8 Tesla (800 MHz ^1H frequency)⁴⁰. These results**

further support our proposal that the exchange between the P and C states explains the intra-burst conductive and non-conductive transitions.

Figure legends (p.32-34)

Fig. 1 The hot spot mutation in the transmembrane region and the NMR analyses of the A111V mutant of KcsA. (B) Overlay of the ^1H - ^{13}C HMQC spectra of the wild type at pH 3.0 (red) and the A111V mutant at pH 3.0 (black) (top), and overlay of the spectra of the wild type at pH 6.5 (blue) and the A111V mutant at pH 3.0 (black) (bottom). The spectra were measured at 45 °C and 18.8 Tesla (800 MHz ^1H frequency), in the presence of 100 mM KCl. The chemical shifts of Leu24 and Val126 in the A111V mutant at pH 3.0 were different from those of the wild type at pH 6.5, because these residues are located proximate to the protonatable residues, His25, Glu118, Glu120, Arg127, and His128, and thereby reflect the local structural differences accompanied by the protonation of KcsA at pH 3.0. Schematic models of the P and C states are shown on the right.

Fig. 2 NMR analyses of the KcsA mutants. (C) Val76 γ 1 methyl signals of the wild type and the V91I, V106I, and A111V mutants, measured at different temperatures. The chemical shifts of the I state (pH 3.0, 25 °C), the P state (pH 3.0, 45 °C), and the C state (pH 6.5, 45 °C) in the wild type are indicated in red, green, and blue boxes, respectively. Schematic models of the equilibrium shifts in the mutants are shown on the right. The spectra were recorded at 11.7 Tesla (500 MHz ^1H frequency, for L36I), 14.1 Tesla (600 MHz ^1H frequency, for L46I), or 18.8 Tesla (800 MHz ^1H frequency, for wild type, V91I, V106I, and A111V).

Fig. 3 Side-chain interactions in the open and closed structures of the HBC-gate. (C) ^1H - ^{13}C HMQC spectra of the wild type and the A28V, A32V, and T33I mutants. The regions for the fingerprint residues, Val76 and Leu59, are shown. The chemical shifts of the P state (pH 3.0, 45 °C) and the C state (pH 6.5, 45 °C) in the wild type are indicated in red and blue boxes, respectively. The spectra were measured at 45°C and 14.1 Tesla (600 MHz ^1H frequency, for A28V and A32V), or 18.8 Tesla (800 MHz ^1H frequency, for T33I), in the presence of 100 mM KCl.

Supplementary information (p. 6)

Supplementary Fig. 5 ^{13}C 1D projections of the V76 γ 1 region (^1H chemical shift range - 0.176 to 0.213 ppm) of the wild type KcsA and its mutants. The spectra were recorded at 18.8 Tesla (^1H frequency 800 MHz), pH 3.0, and 25 $^\circ\text{C}$. The exponential window function (line broadening factor 5 Hz) was applied for the ^{13}C dimension. We fitted the signal line shapes using the Lorentzian function and obtained the linewidths at half height. For the wild type, the impermeable (green) and permeable (red) signals were assumed. For the V91I and V106I mutants, impermeable (green), permeable (red), and closed (blue) signals were assumed. For the A111V mutant, the closed signal (blue) was assumed. The apparent transverse relaxation rates during the t_1 period, $R_{2,app}$, were estimated from the fitted linewidths at half-height, taking the applied line-broadening factor into account.

Point 3: Although the authors do not emphasize this, the channel studied here is solubilized in dodecyl maltoside (DM) micelles. This has been the system of choice for the authors in their previous publications on the KcsA channel. Finding this information was not easy – it is omitted from the Materials & Methods (where it should appear explicitly or in reference to a previous paper) and is only mentioned once in the context of the A111V mutant. DM is a mild detergent with a low CMC and forms micelles that are a convenient membrane-mimicking system for NMR studies. Still, as the authors state themselves (p. 14, Discussion 2nd paragraph) the activity of the K-channel is

influenced by the membrane environment (lipids), and the DM micelle differs from the natural phospholipidic membrane in width and in composition. True, the correlation obtained with single-channel readings gives credence to the results in the DM micelle, but it is plausible that a different choice of surrogate membrane (e.g. lipoprotein nanodiscs) would change the results in the series of sterically-demanding mutations. It might be difficult to draw subtle conclusions regarding the interplay of gates and the influence of transmembrane steric effects in a setting which is less-than-native. It may not be possible to examine this experimentally within the scope of this paper, but this point should be addressed somehow in the text and the membrane-mimicking system chosen stated more clearly. In summary – this is a structure-function study of the KcsA channel with an impressive work content, novelty and importance. Before publication, the paper must be further improved by addressing these questions.

We appreciate the reviewer's critical reading of our manuscript.

We tried to reconstitute KcsA mutants into nanodiscs and observe the NMR spectra; however, it was difficult to quantitatively analyze the conformational equilibrium between the closed and permeable states. This is because: (i) the increase in the apparent molecular weight of the KcsA nanodiscs caused substantial decreases in sensitivity and resolution, and (ii) the relative population of the impermeable state was larger in nanodiscs than in detergent micelles (Imai *et al.*, *J. Biol. Chem.* 287(2), 39634-39641 (2012)), which hampered the NMR observations of the permeable and closed state signals.

As the reviewer pointed out, the conformational landscape of KcsA is indeed affected by differences in the membrane environment. Nevertheless, we still consider our mutant results in detergent micelles to be important for clarifying our understanding of the molecular mechanism of KcsA, according to the following points.

(i) In our previous NMR analyses of KcsA reconstituted in nanodiscs, we observed the same set of conformational substates (the closed, permeable, and impermeable states) in nanodiscs as that observed in DDM micelles. The chemical shift differences in nanodiscs and DDM micelles were relatively small and localized to the region exposed to the surrounding membrane environment (Imai *et al.*, *J. Biol. Chem.* 287(2), 39634-39641). These results suggest that the overall structure of KcsA is not substantially affected by the surrounding membrane environment, and the conformational landscape of KcsA can be described by the same set of conformational substates as that observed in DDM micelles.

(ii) The main conclusion of our manuscript is that a mutation in the transmembrane region

can alter the conformational equilibrium of KcsA and stabilize the conformation with the closed HBC gate. As we compared the effect of each mutation in the common detergent environment throughout the manuscript, it would be reasonable to compare the energetic contribution of each mutation to the conformational landscape of KcsA and expect that the mutations can alter the conformational equilibrium of KcsA to similar extents in native membrane environments, as observed in DDM micelles.

(iii) In our previous NMR analyses of KcsA in DDM micelles, we observed marked equilibrium shifts in the E71A and Y82A mutants, which successfully explained the altered K⁺-channel activity detected in the electrophysiological analyses of KcsA macroscopic currents (Imai *et al.*, *PNAS* 107(14), 6216–6221 (2010)). These results indicate that the NMR investigation of a functionally-altered KcsA mutant in DDM micelles is an effective strategy to understand the gating mechanism of KcsA.

(iv) Although there are few studies that quantitatively compared the conformational equilibrium of membrane proteins in detergent micelles and native membrane environments, we previously characterized the conformational equilibrium of β_2 adrenergic receptor (β_2 AR), a highly plastic alpha-helical membrane protein, in both DDM micelles and nanodiscs (Kofuku *et al.*, *Angew. Chem. Int. Ed.* 53, 13376–13379 (2014), in which the terminology “reconstituted high-density lipoprotein, rHDL” is used to represent nanodiscs). We analyzed the shift in the conformational equilibrium between the two inactive conformations (D and U) and one active conformation (A) from the chemical shift of the M82 methyl signal in the presence of ligands with different pharmacological efficacies, in both DDM micelles and nanodiscs (Supporting figure 5). The relative populations and the exchange rates between these states were different in DDM micelles and nanodiscs; however, the conformational equilibrium of β_2 AR could be described well by the same set of conformational substates, indicating that these three conformational substates of β_2 AR were well preserved and very similar in DDM micelles and nanodiscs. Notably, the ligand-dependent shifts in the conformational equilibrium were also very similar in DDM micelles and nanodiscs (for example, the population of the active conformation, A, increased in the following order: inverse agonist < antagonist < weak partial agonist < partial agonist < full agonist, in both DDM micelles and nanodiscs) (Supporting figure 6).

These results indicate that the energetic contribution of the ligand-binding to the conformational equilibrium of β_2 AR is well-preserved in both DDM micelles and nanodiscs, and NMR investigations of the equilibrium in DDM micelles are still useful to

predict the functional dynamics and activity of β_2 AR in native membrane environments.

[REDACTED]

[REDACTED]

[REDACTED]

Based on these points, we believe our mutant results in DDM micelles are still important in furthering our understanding of the conformational dynamics of KcsA and provide useful insight into the physiological function of KcsA in native membrane environments. To clarify the relevance of our results in detergent micelles to channel functions in native membrane environments, we added the following discussion.

Discussion (p.17, line 21 - p.18, line 12)

In this study, we investigated the structural mechanism underlying the impaired K⁺ channel activity induced by the hot spot mutation. We demonstrated that altered steric contacts in the transmembrane region dramatically affect the conformational equilibrium in the transmembrane HBC gate. We also revealed that KcsA has a similar property to that of the Shaker K⁺ channel and that the introduction of the bulky amino acid in the hot spot region shifts the equilibrium to stabilize the HBC-closed structure, thus leading to the low K⁺-permeation in the mutant. **This structural model is based on our NMR analyses of the KcsA mutants that were conducted in micellar environments, even though the KcsA channel activity is sensitive to the surrounding membrane environments⁴²⁻⁴⁵. However, we previously demonstrated that the three distinct conformational substates of KcsA (P, C, and I states) were well preserved in the native membrane environments, based on NMR analyses of KcsA reconstituted into high-density lipoprotein³⁰. Therefore, it would be reasonable to compare the energetic contribution of each mutation to the conformational landscape of KcsA and expect that the mutations can alter the conformational equilibrium of KcsA in native membrane environments, as observed in the DDM micelles.**

Reviewer #3 (Remarks to the Author):

In this manuscript, Yuta Iwahashi et al. studied a hotspot in potassium channel; researchers have previously discovered the association between the mutations at this position and human diseases. The authors studied this hotspot in a model potassium channel KcsA from bacteria and found out that the A111V mutant cannot enter the activated (P) state at a low pH in contrast to WT. They further demonstrated that the bulky side chain of the V111 prevents the channel from shifting from

the closed-conductive (C) state to the P state, due to steric contact between residues in the transmembrane helix. Therefore, they concluded that the closure at this activation gate (HBC gate) — rather than the inactivated state of the channel — causes the non-conducting behavior in the A111V mutant. They hypothesized that a similar mechanism might explain the behavior of mutants in Shaker and Kir1.1 channels: the change of the side chain in various mutants could affect the equilibrium between the P state and C state and thus the conducting behavior. They also demonstrated that the exchange between the P state and C state causes the intra-burst gating transition observed in the single channel electrophysiology experiments. I find these findings very interesting as they provide a different perspective on the malfunction of potassium channel and single channel electrophysiology. These findings are well supported by their data. I have several questions and some minor suggestions below:

Q1: In this study, Leu59 and V76 peaks were mainly used as fingerprints for indicating the various states of the protein. Even though these residues are coupled to proton binding and are sensitive to both potassium and proton, they are quite distant from the proton binding sites. Have you studied the proton sensor, H25, to see if it gets protonated at a low pH in the A111V mutant, as your group did for WT before? Will the pKa of H25 change in A111V?

We appreciate the reviewer's comment. In our previous analyses, the protonation of His25 was not directly detected. The pH-dependent changes were indirectly monitored by observing the chemical shift changes of the fingerprint methyl groups, Leu59 δ 1 and Val76 γ 1, and then the pH sensor was identified by combining the results from a set of His and Glu mutants and the C-terminally truncated protein (Supporting figure 7). As the structural changes in the transmembrane pore were hampered and the chemical shift changes were not observed for these fingerprint residues in the A111V mutant, it would be technically difficult to adopt the same strategy with this mutant.

[REDACTED]

However, in our NMR analysis of the A111V mutant, some residues exhibited pH-dependent chemical shift changes and we think they reflected the local structural differences induced by the protonation. The methyl groups that showed pH-dependent chemical shift changes in the mutant were located at (i) the cytoplasmic end of the outer transmembrane helix (Leu24) and (ii) the C-terminal helix bundle region (Val126, Leu151, and Leu155), and they exhibited the same or similar chemical shifts as those of the wild type at pH 3.0 (red stars, Supporting figure 8). Regarding (i) the cytoplasmic end of the outer transmembrane helix (Leu24), Leu24 is located next to the pH sensor residue, His25, and is expected to reflect the perturbation of the salt-bridge network formed by His25 under acidic conditions. Regarding (ii) the C-terminal helix bundle region (Val126, Leu151, and Leu155), these chemical shift differences reflected the local structural changes induced by the protonation of the protonatable residues, such as H124, H128, and H145 at the C-terminal region, which were also observed in the wild type KcsA and proved to be uncorrelated to the structural changes at the transmembrane pore region, as evidenced by the difference in $\text{pH}_{1/2}$ (Imai *et al.*, *PNAS* 107(14), 6216–6221 (2010)) (Supporting figure 9).

From these results, we concluded that the protonation did occur in the A111V mutant, but it did not induce the structural changes of the entire transmembrane pore (please also refer to our response to point 2 raised by reviewer 1).

Supporting figure 8 Enlarged view of the Leu/Val region of the ^1H - ^{13}C HMQC spectra of the wild type and the A111V mutant. (A) Enlarged view of the Leu/Val region of overlaid ^1H - ^{13}C HMQC spectra of the wild type at pH 6.5 (blue) and the A111V mutant at pH 3.0 (black). The spectra were measured at 45 °C and 18.8 Tesla (800 MHz ^1H frequency), in the presence of 100 mM KCl. Representative residues observed in the well-resolved region are labeled. The chemical shifts of Leu24, Val126, Leu151, and Leu155 of the wild type at pH 3.0 are indicated as red stars. (B) Mapping of the highlighted residues on the crystal structure of full-length KcsA in the closed state (PDB ID: 3EFF).

[REDACTED]

Even though it may not be necessary, I would recommend more experiments to ensure that HBC is in the closed conformation in A111V at pH3.5?

We appreciate the reviewer's suggestion. We tried to directly observe the residues forming the constriction point of the HBC K⁺-gate (Leu105, Leu110, and Val115) and obtain the distance information of the gate structure, based on the NOE or paramagnetic relaxation enhancement patterns. However, this was technically difficult due to the severe signal overlapping and broadening observed for these residues. The signal broadening presumably reflects the conformational heterogeneity of the salt-bridge network formed between His25, Glu118, Glu120, Arg121, and Arg122, as proposed by Thompson *et al.* (Thompson *et al.*, *PNAS* 105(19), 6900–6905 (2008)), and/or the additional local conformational exchange processes associated with it.

However, it has been widely proposed that the structural changes in the transmembrane

pore region are quite highly cooperative, via an extensive allosteric network formed by Thr74, Ile100, and Phe103 (Imai *et al.*, *PNAS* 107(14), 6216–6221 (2010))(Cuello *et al.*, *Nature* 466, 272–275 (2010)) (Xu *et al.*, *Proc. Natl. Acad. Sci. USA* 114, 8788–8793 (2017)), and the mutation site, Ala111, is distant from these residues ($> 13\text{\AA}$ in the C β -C β distance) and not directly involved in this network. Therefore, the chemical shift pattern of the fingerprint methyl groups in the A111V mutant most likely reflects the formation of a fully or at least partially closed HBC structure.

To clarify these points, we added a more detailed explanation for the pH-dependent spectral changes in the A111V mutant and a supporting figure, as follows.

Results (p.7, line 10- p.8, line 20)

This result indicates that the A111V mutant forms the structure that closely resembles the C state in the wild type, even under pH 3.0 condition. The chemical shifts of the other methyl groups, such as Leu59 δ 2 and Leu86 δ 2, also matched those of the C state of the wild type, indicating that the overall **transmembrane pore** structure of the A111V mutant adopts the structure of the C state observed in the wild type at pH 6.5.

Interestingly, some methyl signals did not follow this trend. These exceptional methyl groups are located at the cytoplasmic end of the outer transmembrane helix (Leu24) and in the C-terminal helix bundle region (Val126, Leu151, and Leu155), and exhibited the same or similar chemical shifts as those of the wild type at pH 3.0 (red stars, Supplementary Fig. 3). In our previous pH titration analyses of KcsA in DDM micelles, we demonstrated that the C-terminal region exhibited a pH-dependent structural transition ($\text{pH}_{1/2} = 6.4 \pm 0.4$) that was not coupled to the pH-dependent gating transition at the transmembrane pore ($\text{pH}_{1/2} = 5.0 \pm 0.3$)¹⁰. In addition, Leu24 is located next to the pH sensor residue, His25, and is expected to be sensitive to the perturbation of the salt-bridge network formed by His25 under acidic conditions^{31,33}. These results indicate that the A111V mutant is likely to be protonated at pH 3.0, which caused the pH-dependent structural transitions at the C-terminal region and the perturbation of the salt-bridge network centered on His25, but could not induce the structural changes of the transmembrane pore, and thus the pore region is forced to adopt the C state conformation, regardless of the pH condition. As the C state represents the non-conductive structure, in which the K⁺-permeation is blocked by the constricted HBC gate, our results indicate that the A111V mutant is a non-conductive mutant whose HBC gate adopts the closed structure even under acidic conditions. It was difficult to directly observe the residues forming the constriction point of the HBC K⁺-gate and obtain the distance information of the gate

structure, due to the signal overlapping and broadening observed in Leu105, Leu110, and Val115, which presumably reflect the conformational heterogeneity of the salt-bridge network formed between His25, Glu118, Glu120, Arg121, and Arg122, as proposed by Thompson *et al.*³³, and/or additional local conformational exchange processes associated with it. However, the structural changes in the transmembrane pore region are quite highly cooperative via an extensive allosteric network formed by Thr74, Ile100, and Phe103^{11–13}, and the mutation site, Ala111, is distant from these residues ($> 13\text{\AA}$ in the C β -C β distance) and not directly involved in this network. Therefore, the chemical shift pattern of the fingerprint methyl groups in the A111V mutant most likely reflects the formation of a fully or at least partially closed HBC structure.

Supplementary information (p. 4)

Supplementary Fig. 3 Enlarged view of the Leu/Val region of the ^1H - ^{13}C HMQC spectra of the wild type and the A111V mutant. (A) Enlarged view of the Leu/Val region of overlaid ^1H - ^{13}C HMQC spectra of the wild type at pH 6.5 (blue) and the A111V mutant at pH 3.0 (black). The spectra were measured at 45 °C and 18.8 Tesla (800 MHz ^1H frequency), in the presence of 100 mM KCl. Representative residues observed in the well-resolved region are labeled. The chemical shifts of Leu24, Val126, Leu151, and Leu155 of the wild type at pH 3.0 are indicated as red stars. (B) Mapping of the highlighted residues on the crystal structure of the full-length KcsA in the closed state (PDB ID: 3EFF)⁶¹.

Q2: In the WT-KcsA, the interactions between charged residues, such as H25, E118, E120, and R121, have been hypothesized to stabilize the closed HBC. As shown in Figure 1B, Leu24 and V126 show chemical shift change due to the low pH in the A111V mutant. It seems to suggest that the salt bridge and hydrogen bonding network of the charged residues are impaired and there is a conformational change at the pH sensor region. Would that mean the HBC gate is probably partially open, but is not large enough for K⁺ to pass through?

We appreciate the reviewer's suggestion. Yes, as we mentioned in the response to Q1, we supposed that the pH-dependent chemical shift changes observed in the A111V mutant reflect the perturbations of the salt-bridge network centered on His25 under acidic conditions. We agree with the reviewer's point, in that we cannot fully exclude the possibility that the HBC gate is partially open but not large enough to allow K⁺ passage, because we did not directly obtain the distance information and only observed the methyl chemical shift differences and signal broadening.

We mentioned the possibility that the HBC gate is not completely closed in the A111V mutant, as follows (the same paragraph we changed in our response to Q1).

Results (p.8, lines 6-20)

As the C state represents the non-conductive structure, in which the K⁺-permeation is blocked by the constricted HBC gate, our results indicate that the A111V mutant is a non-conductive mutant whose HBC gate adopts the closed structure even under acidic conditions. It was difficult to directly observe the residues forming the constriction point of the HBC K⁺-gate and obtain the distance information of the gate structure, due to the signal overlapping and broadening observed in Leu105, Leu110, and Val115, which presumably reflect the conformational heterogeneity of the salt-bridge network formed between His25, Glu118, Glu120, Arg121, and Arg122, as proposed by Thompson *et al.*³³, and/or additional local conformational exchange processes associated with it. However, the structural changes in the transmembrane pore region are quite highly cooperative via an extensive allosteric network formed by Thr74, Ile100, and Phe103¹¹⁻¹³, and the mutation site, Ala111, is distant from these residues (> 13Å in the Cβ-Cβ distance) and not directly involved in this network. Therefore, the chemical shift pattern of the fingerprint methyl groups in the A111V mutant most likely reflects the formation of a fully or at least partially closed HBC structure.

Q3: In Figure 2C panel for V106I at 45 °C, the chemical shift for V76 is in between the values for the P state and C state. As the temperature drops, the peak split into two peaks corresponding to the P state and C state. Does it mean that V76 is in a fast exchange between the P state and C state at 45 °C? But how would the P and C states exchange directly based on the proposed model?

We thank the reviewer for this comment. Yes, it means that the Val76 signal is likely in a fast exchange between the P and C states at 45 °C. If we compare the Val76 signals at 35 °C between the V91I and V106 mutants, the Val76 signal in the V106I mutant is severely broadened and observed in the middle of the permeable and closed chemical shifts, while the signal in the V91I mutant is starting to split into two signals that represent the permeable and closed states. These results suggest that the exchange rate is faster in the V106I mutant as compared to the V91I mutant, and that the exchange in the V106I mutant is in the intermediate exchange regime, while that in the V91I mutant is skewed to the slow exchange regime. This notion is also supported by the result that the Val76 signal was more severely broadened in the ¹³C dimension at 25 °C in the V106I mutant, as compared to the V91I mutant (Supplementary Fig. 4) (please refer to our response to Q5 below for more details about the line-shape analyses).

Although we cannot fully describe a detailed exchange pathway from our thermodynamic analyses, we expect that a direct transition exists between the P and C states and between the P and I states ($C \rightleftharpoons P \rightleftharpoons I$), and a direct transition occurs between the P and C states during the burst period. This is because the P and C states share a common structural feature at the SF gate (open SF gate), while the P and I states share a common structural feature at the HBC gate (open HBC gate). If we assume the direct transition between the C and I states, then we have to assume that structural changes occur at both the HBC and SF gates, which presumably are energetically unfavorable. Moreover, we expect the free energy barrier for the structural rearrangements of the HBC gate to be relatively low, based on the observation that the pH-dependent activation/deactivation process occurs with a timescale on the order of 10-100 ms (please refer to our response to Q5 for details). Therefore, it would be reasonable to assume that the direct transition occurs between the C and P states in our model.

To clarify this point, we added an explanation of the exchange regime of the V106I mutant, as follows.

Results (p.10, lines 5-19)

Interestingly, the signals from all three of the states in these mutants were observed below 40 °C, and their relative populations changed in a temperature-dependent manner. These results suggest that these KcsA mutants exist in a similar conformational equilibrium between the three states to that observed in the wild type, and the C state is more stabilized with the V91I and V106I mutations under the acidic conditions, as compared to the wild type. The observed temperature-dependent spectral changes also indicated that the exchange kinetics between these states was different between the mutants. In the V91I mutant, the signals from all three states were separately observed, while in the V106I mutant, the signals from the P and C states were merged to give a single broad peak at temperatures above 35 °C, strongly suggesting that the exchange rate between the P and C states is faster in the V106I mutant than in the V91I mutant. Notably, in the A111V mutant, the P and I states were not observed between 25 and 45 °C, demonstrating that the introduction of the bulky amino acid at the hot spot position greatly alters the conformational equilibrium and forces KcsA to adopt exclusively the C state with the closed HBC gate.

Q4: How many single channel recording current traces were used for the analysis? It is worth noting that different modal conducting behaviors were observed in WT-KcsA (Chakrapani S, Cordero-Morales JF, Perozo E. A quantitative description of KcsA gating II: single-channel currents. J Gen Physiol. 2007;130(5):479-496. doi:10.1085/jgp.200709844)

We thank the reviewer for comment. We analyzed multiple single-channel recordings for the wild type and each mutant, except for the A111V mutant, since its open probability was very low and the opening event was rarely observed.

Regarding the different modal conducting behaviors, we observed only the high- P_o mode with the open probability of ~ 0.82 , as shown in Fig. 4, which is consistent with the previous observation that this high- P_o mode is the most prevalent gating mode (Chakrapani *et al.*, *J. Gen. Physiol.* 130(5), 479–496 (2007), Chakrapani *et al.*, *Nat. Struct. Mol. Biol.* 18(1), 67–74 (2011)). The reason why we did not observe the flicker mode or the low- P_o mode in our recordings can be attributed to the differences in the experimental conditions (azolectin lipid and MOPS buffer were used in the previous studies by Chakrapani, and diphytanoyl phosphatidylcholine and potassium phosphate buffer were used in our study). NMR spectra generally represent a population-weighted average of conformational ensembles of proteins; therefore, it is reasonable to assume that the NMR spectra of the wild type KcsA represent the conformation that is responsible for the high- P_o mode of gating behavior.

To clarify this point, we added more details about the single-channel recordings as follows.

Results (p.12, lines 14-22)

In the wild type KcsA, the populations of the conductive and the non-conductive states during the burst period were calculated to be 84 % and 16 %, respectively (Fig. 4A top, red and blue dotted lines). In the previous reports by Chakrapani *et al.*, three different modal conducting behaviors were observed during the burst period in the single-channel recordings of the wild type KcsA: the high- P_o mode with the open probability of 82 %, the flicker mode with the open probability of 40 %, and the low- P_o mode with the open probability of 16 %^{38,39}. In our single channel analyses, only the high- P_o mode was observed in the wild type recordings, which is consistent with the previous observation that the high- P_o mode is the most prevalent gating mode of the wild type KcsA.

Methods (p. 22, line 22- p.23, line 8)

The single-channel analyses of KcsA were performed by using a Port-a-Patch system (Nanion Technologies). The planar lipid bilayers were formed by adding GUVs to the chamber under negative pressure, and 1 μ L of DDM-solubilized KcsA (50 μ M) was directly added to the bilayer for reconstitution. The internal and external solutions contained 20 mM potassium phosphate (pH 6.5/3.0), 100 mM KCl, and the constant transmembrane voltage was set to -200 mV. All traces were recorded with a 40 kHz sampling rate at room temperature (around 18 °C). Multiple gating events were analyzed for the wild type and the mutant KcsA proteins ($n=2-3$), except for the A111V mutant in which the opening event was rarely observed. Data were analyzed with QuB (<https://qub.mandelics.com/>)⁶¹.

Q5: The conducting and non-conducting dwell times in the intra-burst are on the millisecond time scale. Would you expect to see line broadening of the NMR peaks due to this millisecond exchange process?

We appreciate the reviewer's careful reading of our manuscript. As the reviewer pointed out, we did observe the line-broadening effects in the V76 γ 1 methyl signals in the V91I and V106I mutants. To clarify this point, we added the 1D projections of V76 γ 1 in the ¹³C dimension in Supplementary Fig. 5 (please also refer to our response to point 2 from reviewer 2).

We focused on the V76 γ 1 methyl signal of the permeable state and contribution of the exchange between the permeable (major) and closed (minor) states to the exchange-

induced line-broadenings, as we previously found that the exchange between the impermeable state was very slow ($k_{\text{ex}} = 1.4 \text{ s}^{-1}$ at $40 \text{ }^{\circ}\text{C}$) (Imai *et al.*, *J. Biol. Chem.* 287(2), 39634-39641). From the comparison of linewidths between the wild type and the V91I mutant, the exchange contributions in the ^{13}C linewidth were estimated to be around 60 s^{-1} , assuming that the exchange contribution in the wild type is minimal because of the small population of the closed state (minor state). If we assumed the ^{13}C chemical shift difference of 0.45 ppm, the minor state population of 30 %, and the exchange rate of 160 s^{-1} , then the exchange contribution in the ^{13}C linewidth at 800 MHz was estimated to be 46.9 s^{-1} , which was agreeably consistent with the experimentally observed line-broadening. We added a supplementary figure and an explanation for the exchange broadenings observed in the V76 γ 1 methyl signal. We also stated the magnetic field strength in the figure legends, as follows.

Results (p.14, line 12 - p.15, line 4)

We analyzed the intra-burst gating kinetics of the V91I mutant and calculated the mean dwell times of the conductive and non-conductive states within the burst period (Supplementary Fig. 4). The mean dwell times of the conductive and non-conductive states were 7.9 and 27.0 ms, respectively, and correspond to an exchange rate of 160 s^{-1} assuming two-state exchange. These kinetic parameters are consistent with the NMR observation that the exchange between the P and C states was in a slow-exchange regime with significant exchange-induced line broadening effects, and cannot be explained by the exchange between the P and I states, as we previously demonstrated that this exchange process is much slower, with a k_{ex} of 1.4 s^{-1} at $40 \text{ }^{\circ}\text{C}$. In the NMR spectrum of the V91I mutant at $25 \text{ }^{\circ}\text{C}$, extra line broadening effects on the order of several tens of seconds were observed for the V76 γ 1 methyl signals in the ^{13}C dimension, as compared to the wild type (Supplementary Fig. 5), consistent with the expected exchange contribution ($= 46.9 \text{ s}^{-1}$) assuming the exchange rate of 160 s^{-1} , the ^{13}C chemical shift difference of 0.45 ppm, and the major and minor state populations of 70 % and 30 % at 18.8 Tesla (800 MHz ^1H frequency)⁴⁰. These results further support our proposal that the exchange between the P and C states explains the intra-burst conductive and non-conductive transitions.

Supplementary information (p. 6)

Supplementary Fig. 5 ^{13}C 1D projections of the V76 γ 1 region (^1H chemical shift range - 0.176 to 0.213 ppm) of the wild type KcsA and its mutants. The spectra were recorded at 18.8 Tesla (^1H frequency 800 MHz), pH 3.0, and 25 $^\circ\text{C}$. The exponential window function (line broadening factor 5 Hz) was applied for the ^{13}C dimension. We fitted the signal line shapes using the Lorentzian function and obtained the linewidths at half height. For the wild type, the impermeable (green) and permeable (red) signals were assumed. For the V91I and V106I mutants, impermeable (green), permeable (red), and closed (blue) signals were assumed. For the A111V mutant, the closed signal (blue) was assumed. The apparent transverse relaxation rates during the t_1 period, $R_{2, \text{app}}$, were estimated from the fitted linewidths at half-height, taking the applied line-broadening factor into account.

Even though it is known that the activation of the HBC gate is quite fast (millisecond time scale), do you think the transition between the closed and open conformations of HBC is also fast?

We thank the reviewer for this comment. Our previous K^+ titration experiments (Imai *et al.*, *PNAS* 107(14), 6216–6221 (2010)) indicated that the SF gate of the P and C states represents a K^+ -bound, conductive conformation, as observed in the crystal structure in the presence of K^+ ; therefore, we think the structural differences between the P and C

conformations mainly occur at the HBC gate, which is also responsible for the pH-dependent activation/deactivation process. As the reviewer pointed out, the pH-dependent closed to open transition at the HBC gate is quite fast, on the order of 10-100 ms. We point out that the reverse process, the open to closed transition at the HBC gate, is also fast as demonstrated by the pH-jump experiments observing the KcsA macroscopic current (Chakrapani *et al.*, *J. Gen. Physiol.* 130(5), 465–478 (2007)). Chakrapani *et al.* analyzed the pH-dependent deactivation process of KcsA by rapidly changing the pH from 3.0 to 8.0 at the peak of the current, and then measuring the rate of channel closing. Their results showed that the deactivation also occurred within a timescale on the order of tens of milliseconds (Supporting figure 10).

These results support our notion that the free energy barrier for the structural rearrangements of the HBC gate is relatively small, and that the transition between the closed and permeable conformations accompanying the rearrangement of the HBC gate is expected to be quite fast and on the millisecond timescale.

[REDACTED]

To clarify this point, we rewrote the Discussion section as follows.

Results (p.19, lines 10-15)

These exchange kinetics profiles are in good agreement with the results of the electrophysiological measurements of the KcsA macroscopic currents, **in which the pH-dependent activation or deactivation process, involving the structural changes at the HBC gate, occurs on a 10-10² millisecond time scale**, whereas the subsequent inactivation process involving the SF gate closure occurs on a 1-10 second time scale^{27,32,41,51}.

Minor comments

1. Since Leu59 and Val76 were mainly investigated in the paper, it is probably better to also highlight their positions in the main figure such as in the crystal structure in Figure 1.

We thank the reviewer for the suggestion. We highlighted the positions of Leu59 and Val76 on the crystal structure in Fig. 1A.

Figures (p. 32)

Fig. 1 The hot spot mutation in the transmembrane region and the NMR analyses of

the A111V mutant of KcsA. (A) Sequence alignment of *Streptomyces lividans* KcsA (Uniprot ID: P0A334), *Homo sapiens* Kv1.1 (Uniprot ID: Q09470), and *Drosophila melanogaster* Shaker Kv channel (Uniprot ID: P08510). The hot spot positions in the inner transmembrane helix are highlighted in magenta. The hot spot position of KcsA, Ala111, is indicated in the crystal structure of KcsA (PDB ID: 1K4C). Only two facing subunits are shown for clarity. **The positions of the fingerprint methyl groups, Val76 γ 1 and Leu59 δ 1, are highlighted in orange.** (B) Overlay of the ^1H - ^{13}C HMQC spectra of the wild type at pH 3.0 (red) and the A111V mutant at pH 3.0 (black) (top), and overlay of the spectra of the wild type at pH 6.5 (blue) and the A111V mutant at pH 3.0 (black) (bottom). The spectra were measured at 45 °C and 18.8 Tesla (800 MHz ^1H frequency), in the presence of 100 mM KCl. The chemical shifts of Leu24 and Val126 in the A111V mutant at pH 3.0 were different from those of the wild type at pH 6.5, because these residues are located proximate to the protonatable residues, His25, Glu118, Glu120, Arg127, and His128, and thereby reflect the local structural differences accompanied by the protonation of KcsA at pH 3.0. Schematic models of the P and C states are shown on the right.

2. *The conformation of KcsA is dependent on potassium concentration. Were all the experiments done at 100 mM K⁺? I think you need to mention the [K⁺] where you discussed the sample conditions in the main text, at least at the very beginning.*

We thank the reviewer for this suggestion. All of the experiments in this manuscript were conducted in the presence of 100 mM KCl. To clarify this point, we added explanations in the main text and the Materials and Methods section, as follows.

Results (p. 6, line 24 - p. 7, line 4)

We compared the ^1H - ^{13}C heteronuclear multiple quantum coherence (HMQC) spectra of the wild type and the A111V mutant solubilized in *n*-dodecyl- β -D-maltopyranoside (DDM) at pH 3.0 and 45 °C **in the presence of 100 mM KCl**, using the uniformly deuterated and Ile δ 1, Leu δ 1/ δ 2, and Val γ 1/ γ 2 selectively methyl- $^1\text{H}_3$ - ^{13}C labeled samples (Fig. 1B **and Supplementary Fig. 2**).

Results (p. 9, lines 10-13)

We observed HMQC spectra of 25 mutants (V34I, L35I, L36I, V37I, V39I, L40I, L41I, L46I, V48I, L49I, L66I, V70I, L81I, V84I, L86I, L90I, V91I, V93I, V94I, V95I, V97I, L105I, V106I, L110I, and V115I) at pH 3.0 and 45 °C **in the presence of 100 mM KCl**, and

compared the conformational states of these mutants with that of the wild type.

Methods (p. 22, lines 5-7)

All NMR spectra were recorded in buffer containing 20 mM potassium phosphate (pH 3.0 or 6.5), 100 mM KCl, and ~ 5 mM DDM in 100% D₂O. The pH values in D₂O were calibrated by adding 0.4 pH unit to the reading on the pH meter⁵⁹.

3. *Is the protein in its full-length or truncated form? It would be nice to clarify this.*

We used the full-length KcsA (residues 1-160) in all of the experiments presented in the manuscript. To clarify this point, we added an explanation in the Materials and Methods section, as follows.

Methods (p. 21, lines 7-10)

The plasmid for the expression of the N-terminally decahistidine tagged wild type KcsA (residues 1-160) was constructed as described previously^{10,30,31}. In this study, the C-terminal region of KcsA was not cleaved by chymotrypsin, and thus the full-length KcsA protein was analyzed.

4. *In the figure caption of Figure 1, “L25” is a typo. It should be “L24”.*

We thank the reviewer for pointing out our mistake. We corrected the caption of Fig. 1.

Fig.1 legend (p. 32)

The spectra were measured at 45 °C and 18.8 Tesla (800 MHz ¹H frequency), in the presence of 100 mM KCl. The chemical shifts of Leu24 and Val126 in the A111V mutant at pH 3.0 were different from those of the wild type at pH 6.5, because these residues are located proximate to the protonatable residues, His25, Glu118, Glu120, Arg127, and His128, and thereby reflect the local structural differences accompanied by the protonation of KcsA at pH 3.0. Schematic models of the P and C states are shown on the right.

REVIEWERS' COMMENTS

Reviewer #1 (Remarks to the Author):

After reviewing the revised manuscript from Iwahashi et al. I believe my major concerns were addressed. I believe this manuscript is acceptable in its current form.

Reviewer #2 (Remarks to the Author):

The authors have thoroughly addressed my concerns raised in the review of the original manuscript. In my opinion it is suitable for publication.

Reviewer #3 (Remarks to the Author):

Thank you for the authors' response. I am generally satisfied with their answers. But I have a few more questions.

1. In the S8 figure, could you add assignment labels for A111V pH3 spectrum (the black spectrum)? Where is the L24 peak for A111V at pH3? For the residues like L24 in the WT, the chemical shift could come from two parts: perturbation due to protonation of H25; structural change due to opening of HBC gate, if we break down the protonation and HBC gate opening into two steps. Would L24 show a chemical shift different from that of WT at both neutral and acidic pH for A111V pH 3 sample?

2. Are the protein studied specifically labeled? I see that many peaks are not assigned in like the S8 figure. Would you provide more information on how you assign these peaks?

3. The correlation between electrophysiology and NMR measurements is very interesting. But it would be more convincing that more single channel recording traces are analyzed. 2-3 traces seem a little short-handed.

Point-by-point responses to reviewers

Reviewer #1 (Remarks to the Author):

After reviewing the revised manuscript from Iwahashi et al. I believe my major concerns were addressed. I believe this manuscript is acceptable in its current form.

We greatly appreciate the reviewer for the constructive comments and suggestions on our work. We believe that these comments and suggestions have significantly improved our manuscript.

Reviewer #2 (Remarks to the Author):

The authors have thoroughly addressed my concerns raised in the review of the original manuscript. In my opinion it is suitable for publication.

We greatly appreciate the reviewer for the constructive comments and suggestions on our work. We believe that these comments and suggestions have significantly improved our manuscript.

Reviewer #3 (Remarks to the Author):

Thank you for the authors' response. I am generally satisfied with their answers. But I have a few more questions.

1. In the S8 figure, could you add assignment labels for A111V pH3 spectrum (the black spectrum)? Where is the L24 peak for A111V at pH3? For the residues like L24 in the WT, the chemical shift could come from two parts: perturbation due to protonation of H25; structural change due to opening of HBC gate, if we break down the protonation and HBC gate opening into two steps. Would L24 show a chemical shift different from that of WT at both neutral and acidic pH for A111V pH 3 sample?

We appreciate the reviewer's positive comments and careful reading of our manuscript. According to the reviewer's suggestion, we added the assignment labels for the spectrum of the A111V mutant at pH 3.0 (Fig. S3, black spectrum). Although the L24 δ 1 methyl signal was difficult to assign in the A111V mutant, due to the severe signal overlap, we indicated the position of the L24 δ 2 methyl signal in the spectrum. The chemical shift value of L24 δ 2 was closer to that of the wild type at pH 3.0. As the other methyl signals from the transmembrane pore showed the similar chemical shifts to those of the wild type at pH 6.5

(the closed state), the methyl chemical shift of L2482 presumably reflects the local structural differences around the pH-sensor residues induced by the protonation of His25.

Supplementary Information (page 4)

Supplementary Fig. 3 Enlarged view of the Leu/Val region of the ^1H - ^{13}C HMQC spectra of the wild type and the A111V mutant. (A) Enlarged view of the Leu/Val region of overlaid ^1H - ^{13}C HMQC spectra of the wild type at pH 6.5 (blue) and the A111V mutant at pH 3.0 (black). The spectra were measured at 45 °C and 18.8 Tesla (800 MHz ^1H frequency), in the presence of 100 mM KCl. Representative residues observed in the well-resolved region are labeled (black: the A111V mutant at pH 3.0, blue: wild type at pH 6.5, red: wild type at pH 3.0). The chemical shifts of Leu24, Val126, Leu151, and Leu155 of the wild type at pH 3.0 are indicated as red stars. (B) Mapping of the highlighted residues on the crystal structure of the full-length KcsA in the closed state (PDB ID: 3EFF)⁶¹.

As the reviewer pointed out, it would be more informative, if we can separate the effect of the protonation from that of the HBC gate opening on the chemical shift perturbation pattern, however, these two steps are very highly correlated and technically difficult to separate these two contributions. For example, if we perturb the protonation of KcsA by introducing the mutation to the pH-sensor residue, H25, the HBC gate opening does not occur under acidic conditions as we previously demonstrated in our NMR analyses (Imai *et al.*, *PNAS* 107(14), 6216–6221 (2010)). Also, in order to observe the HBC-open structure

under basic conditions, the deletion of the N-terminal and the introduction of multiple mutations are needed (Δ 1-20, H25Q, R117Q, E120Q, R121Q, R122Q and H124Q, known as KcsA-OM), which would greatly alter the local structure around L24 and hamper the interpretation of chemical shift perturbation patterns (Cuello *et al.*, *FEBS Lett.*, 584, 6, 1133-1132010).

If we could observe the NMR signals of L24 that solely reflects the structure of the HBC gate, we expect the chemical shift of L24 to be very similar to that of the wild type at pH 6.5, as we observed in the other methyl residues that are located apart from H25.

2. Are the protein studied specifically labeled? I see that many peaks are not assigned in like the S8 figure. Would you provide more information on how you assign these peaks?

We thank the reviewer for the comment. The proteins used in the NMR analyses were methyl- ^{13}C -labeled at Ile δ 1, Leu δ 1/ δ 2, and Val γ 1/ γ 2 positions in a uniformly deuterated background. The assignments were established by site-directed mutagenesis as described previously (Imai *et al.*, *PNAS* 107(14), 6216–6221 (2010)). All Leu, Val and Ile residues in KcsA were mutated to Ile (for Leu/Val) or Val (for Ile) one by one, and the NMR spectra of each mutant were compared to the wild-type spectra obtained at pHs 6.7 and 3.2. The stereospecific assignments were obtained by recording the ^1H - ^{13}C HMQC spectrum of the stereospecifically Leu δ 2 and Val γ 2 ^{13}C -labeled KcsA. In addition, we analyzed methyl-methyl NOE cross peaks in the ^{13}C -edited NOESY HMQC spectrum and confirmed that the observed NOEs were consistent with the crystal structure.

To clarify this point, we added the description of the assignment process in the material and method section as follows.

Material and Method (page 23)

The assignments were established by site-directed mutagenesis using the wild type KcsA protein¹⁰, and the assignments were transferred to the KcsA mutants. Briefly, all Leu, Val and Ile residues in KcsA were mutated to Ile (for Leu/Val) or Val (for Ile) one by one, and the ^1H - ^{13}C HMQC spectra of each mutant were compared to the wild-type spectra. The stereospecific assignments were obtained by recording the ^1H - ^{13}C HMQC spectrum of the uniformly deuterated and selectively Leu δ 2 and Val γ 2 methyl- $^1\text{H}_3$ - ^{13}C labeled sample⁵⁸.

3. The correlation between electrophysiology and NMR measurements is very interesting. But it would be more convincing that more single channel recording traces are analyzed. 2-3 traces seem a little short-handed.

We appreciate the reviewer's critical reading of our manuscript. In 2-3 single channel trace, we could observe many opening and closing events and statistically analyzed these data to estimate the open probability and the gating kinetics. For example, in the V91I traces, we analyzed more than 600 open and closed dwell times, which yielded open/closed distributions that were well-fitted by assuming standard Gaussian distributions (Fig. 4B) and also reasonable open and closed dwell time distributions as expected from probability distribution function assuming the two-state gating model (supplementary Fig.4). Therefore, we believe the amount of the of data points was sufficient to faithfully estimate the gating kinetics during the burst period.

We have to admit that the number of the data points of the V106I mutant was smaller than the wild type and the V91I mutant due to the instability of the patch containing the V106I protein, which was about 20% of that used in the analyses of the V91I mutant. In order to evaluate the effect of the number of the data points on the calculated open probability, we conducted Monte Carlo analyses, using the V91I mutant data set. In this analysis, we randomly omitted the experimental data points from the V91I mutant data set to produce the simulated data set with the reduced number of the data points (2, 5, 10, 20, 30, 50 % of the original data), and then calculated the open probabilities from these reduced data sets. Supporting figure below shows the plot of the calculated open probability versus the number of the data points, in which the error bars represent the 2-standard deviation of results of the 500 repeated random samplings. The calculated open probabilities were very robust and 20% of the data point was enough to faithfully estimate the open probability, and the error caused by the reduced sampling was smaller than the experimental error. Thus, we concluded that the open probability of the V106I mutant was not obscured by the smaller number of the datapoints.

Supporting figure Monte Carlo error analysis to evaluate the effect of the reduced sampling density on the calculated open probability.

The single channel traces of the V91I mutant were used. The number of the data points were randomly selected to produce the simulated datapoints containing 2, 5, 10, 20, 30, 50 % of the original data points, and then the open probabilities were calculated from these reduced data sets. The open probabilities were calculated based on a half-amplitude threshold. The error bars represent the 2-standard deviation of results of the 500 repeated random samplings.